## Registered report

psychology/environmental science

climate change, activism, collective action, pro-environmental behaviour, prosocial behaviour

**Author for correspondence:**
Anna Castiglione
e-mail: acastigl@ucsd.edu

# Discovering the psychological building blocks underlying climate action—a longitudinal study of real-world activism

Anna Castiglione[1], Cameron Brick[2], Stefanie Holden[1], Ella Miles-Urdan[1] and Adam R. Aron[1]

[1]Department of Psychology, University of California, San Diego, CA, USA
[2]Department of Psychology, University of Amsterdam, Amsterdam, The Netherlands

 AC, 0000-0001-6916-6760; CB, 0000-0002-7174-8193

We are in a climate emergency. Because governments are reacting too slowly, grassroots collective action is key. Understanding the psychological factors underpinning engagement can facilitate the growth of such collective action. Yet, previous research in psychology rarely provided causal evidence for which factors trigger action, lacked focus on the climate crisis, was mostly self-reported behaviour or intentions rather than objective measures, and was mostly cross-sectional rather than longitudinal. Here we conducted a longitudinal study on the effectiveness of a 12-week video intervention designed to increase psychological predictors of collective action. The intervention boosted affective engagement, collective efficacy, and self-efficacy, but did not increase observed attendance of activism events. Interviews suggested that Zoom fatigue and the online study design undercut the social interaction participants wanted in order to join events. However, a smaller in-person replication did not increase activism either. Debriefings suggested that the replication participants were primarily motivated by payment and lacked time or resources for more engagement. These results highlight the crucial importance of going beyond measures of self-reported attitudes or intentions to objectively measuring activism behaviours and showing the difficulty of fostering event attendance.

## 1. Introduction

Since the industrial revolution in the early 1800s, humans have added carbon dioxide ($CO_2$) and other greenhouse gases to the

atmosphere causing global heating of about 1°C, and this is now increasing exponentially [1,2]. In 2018, the Intergovernmental Panel on Climate Change (IPCC), backed by the world's governments, urged a reduction of emissions of 45% by 2030 from 2010 levels to keep heating to relatively safe levels [3]. This would require rapid and far-reaching transitions in land use, energy systems, buildings and transportation [4]. However, after 30 years of international talks, there is no binding treaty on emission reductions, nor are most governments doing enough locally [5]. An alternative avenue is for collective action movements to grow at the grassroots, which will eventually lead to political pressure [6]. While such groups do already exist (e.g. Extinction Rebellion, the Sunrise Movement, 350.org and Fridays for Future), the overall membership is still small. In this study, we aim to better understand the psychological factors that drive individuals to join collective action.

Research in this area has mostly focused on broader environmental concerns, with only a few studies specifically related to climate change. Based on a non-systematic review [7], we identified a dozen psychological factors that relate to self-reported environmental activism, where this includes joining events organized by ecologically oriented groups, and engaging in environmental education and leadership behaviors (such as circulation of petitions, raising awareness about climate issues, outreach and community organizing; [8–13]). The findings emerged in diverse geographical locations (although predominantly white, developed western nations were studied), ranging across Germany [12], Austria [14], the USA [15,16], Australia [17] and other countries such as Argentina, Chile, South Korea, Mexico, Russia, Turkey and others [15]. We identified psychological factors including affective engagement (i.e. anxiety, sense of threat, worry and concern about the environment and ecological extinction), worldviews, i.e. egalitarian [15,16], nature-loving [18] and anti-consumerist beliefs [19], collective efficacy (i.e. the perception that the actions of one's group can have an impact) and social norms (i.e. the perceived approval of one's engagement in activism from people in one's social circle; [12,14,17]).

While this literature provides important clues about these key psychological factors, the studies are subject to four key limitations. First, there is a *lack of causal evidence*. Almost all studies were cross-sectional (i.e. the relation between self-reports of psychological factors and people's activism was only measured at one time point). For example, in a nationally representative survey conducted in 2008 [16], participants rated their risk perceptions and collective efficacy about climate change. They then self-reported their past climate actions and their future intentions (e.g. contacting elected officials and attending rallies or meetings). Cross-sectional designs with one time point are appropriate and informative, but only provide weak evidence for causation, which could for example be driven by an unknown third variable.

Second, there is substantial *noise and bias in self-report measures of activism behaviour*. Many studies measured recall for distant memories of activist behaviour. Measurement noise increases when the recalled events are further back in time, and when the measures do not specify the time or frequency [13,16]. Some studies also used unclear categorizations of what kind of activism was done and at what intensity [20]. Indeed, evidence is emerging that self-reported pro-environmental behaviour is biased due to noisy recall [20,21], and is also influenced by people exaggerating their green identities when they are observed [22,23].

Third, there is a *lack of climate focus*. Most previous studies on environmental activism were not specifically about climate change. The climate crisis is different from other environmental issues such as plastic waste and local industrial pollution. The climate concern requires acting on a threat that is often invisible, gradual, and long-term [24–26]. It is also global in the sense that local actions are only ultimately effective in concert with massive changes elsewhere. The climate threat is also unusual in that it is so widely distributed as to almost defy comprehension: it is leading to cascading problems across such diverse areas as agriculture and biodiversity, socio-political stability and human health [3]. These features indicate unique barriers to participation in climate action, which may require studies specifically on climate activism.

Fourth, as with any research project, there are also specific *methodological weaknesses* in this area. Almost no studies verified that participants believed key facts about climate change (cf. [16]). Without that, it is difficult to interpret the role of a factor like collective efficacy in relation to activism due to a potential for collider bias [27]: e.g. a person might report high collective efficacy but not engage in climate activism because they do not believe in anthropogenic climate change, which can obscure the role of collective efficacy in a target population believing in climate change. Additionally, many of these studies only measured hypothetical intentions as their outcome variable [10,11,18,19], rather than objective measures of behaviour. Self-reported pro-environmental behaviours only correlate about $r_{17} = 0.46$ with objective measures of the same behaviours [21].

## 1.1. Current study

We designed an online study to address some of these weaknesses. The aim of this study was not to test any particular theory, but rather to develop a novel methodology that clearly identifies which particular psychological factors may cause climate activism. To do this, we introduced two novel tools: (1) an intensive video intervention attempting to boost 11 psychological factors, and (2) a behaviour-tracking methodology to measure actual participation in Zoom activism events organized by two climate organizations.

Specifically, we conducted a three-month longitudinal study aimed at boosting psychological factors such as collective efficacy and social norms. Although our review of the literature revealed more than a dozen possible psychological factors, we settled on just 11. These both had the most correlational evidence with activism behaviour, and fit our intuitions as climate activists of being relevant. The main goal of our study was to see which psychological factors changed in response to the intervention and to identify which of those changes predict shifts in behaviour. The main measure of activism behaviour in this study was event participation (objectively recorded). We embedded a study team member in the UCSD Green New Deal and SD350 climate groups and verified participation in events held by both organizations. Additionally, self-reported environmental education and leadership behaviour was collected, along with other pro-environmental behaviours relevant to climate change. Finally, a semi-structured interview based on the confirmatory results provided qualitative data about activists' own experiences and perceived barriers. All anonymized data, code and materials are available at the Open Science Framework: https://osf.io/38vkz/?view_only=fbc1a81292a5404c9418c83c5fa06c93.

# 2. Study 1

## 2.1. Methods

Please see the Supplementary Materials for the screening questions, baseline and follow-up surveys, the semi-structured interview schedule and the 12 videos and comprehension questions.

### 2.1.1. Sample size

We calculated the required sample based on an effect size of $f^2 = 0.15$ for the regression analysis of hypothesis H3 (see below). This smallest effect size of interest was determined by aggregating across the literature of the 11 psychological factors. The R package *pwr* was used to perform a power analysis for a linear regression with alpha = 0.05, power = 0.95, effect size $f^2 = 0.15$ and numerator d.f. = 5. This power analysis yielded $N = 143$. Based on our earlier experience recruiting student activists through classes and educational materials, we anticipated 30% of participants would join at least one activism event, and half of these participants would develop an enduring commitment. Further, we expected about 10% of participants to drop out during the study. Therefore, we planned to recruit $N = 160$ for the main experimental group.

#### 2.1.1.1. Outliers
Outliers were not relevant to our outcome measures as they are bounded by limited response options or limited available events.

### 2.1.2. Design

We used a single-cohort pre-post design with a pre-intervention baseline phase.

### 2.1.3. Recruitment

We recruited 170 UCSD students between 18 and 38 years of age through online department platforms and by flyers posted on the campus of UC San Diego. The inclusion criteria were that participants were enrolled at UCSD, that they believed in anthropogenic global heating and that they had no or low prior engagement in climate activism (see Screening Survey). The drop-out rate was higher than expected: 30% of the participants dropped out before the study began, and 13% dropped out during the three-month study period, so the final sample was $N = 96$ (22 males, 72 females and 2 unspecified). 20% of the

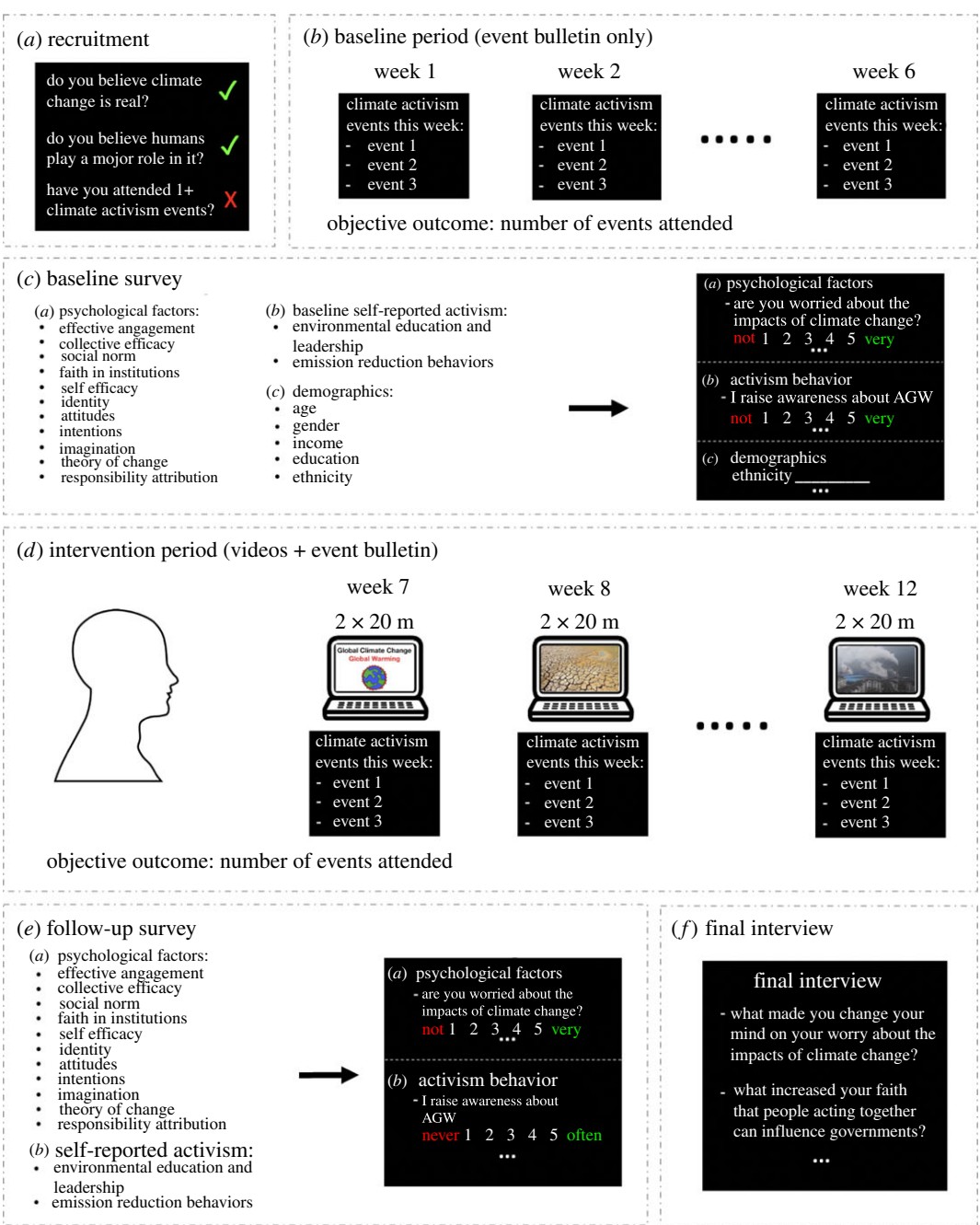

**Figure 1.** *Experimental Design. Note.* (*a*) Screening for beliefs in anthropogenic climate change and attendance to climate activism events. (*b*) Six-week baseline period of receiving climate activism event bulletins every week. Attendance was objectively recorded. (*c*) Baseline survey. (*d*) Six-week intervention period with 20-minute videos, twice per week, and the event bulletins. (*e*) After three months, follow-up survey of the same psychological factors and activism behaviors. (*f*) Some participants completed a semi-structured interview.

participants were Caucasian, 51% Asian, 18% Hispanic/Latino and 11% had other ethnic/racial backgrounds. 75% of the participants identified as Democrats, 5% as Republicans and 20% as Other.

### 2.1.4. Screening survey

Participants underwent an initial screening for age (18–40) and then beliefs in anthropogenic global heating through two questions: 'Regardless of its cause, I am certain that climate change is actually occurring' (yes/no), and 'Human activities are a significant cause of climate change' (yes/no) (figure 1*a*). Participants answering no to either were excluded.

A concern in our design was that during the study some participants might attend events of climate organizations other than UCSD Green New Deal and SD350 and that we would not be able to tell. To limit this possibility, we ensured people had no history of activism. Being engaged in activism outside our study would have made it more likely for a participant to attend an activist event that we could not track during the study. Therefore, we asked everyone at recruitment if they had ever attended an event organized by an environmental organization (such as a protest). Those answering yes were then asked how many times they attended such an event (once, more than once). 15 participants responding 'more than once' were excluded from the study.

The remaining participants read an instruction sheet and approved a consent form about the timeline of the study and their responsibilities over the three-month period. This included rules for payment penalties based on not completing the tasks. Then, each week for 12 weeks, participants received two emails with links to a Qualtrics survey (one link each at the beginning and the end of the week).

### 2.1.5. Baseline period (six weeks)

For the first six weeks, the Qualtrics surveys received by the participants contained a 'climate events bulletin', consisting of a list of events happening in the next 3–4 days, held by our partner climate organizations (San Diego 350 and UCSD Green New Deal) (figure 1*b*). At the top of the bulletin, participants were reminded that while they were required to scroll to the end of the list, they were not required to attend these events (there was no financial penalty for not attending). To ensure that participants read the events bulletin, they were asked two attention-check questions at the end of each bulletin about the events. Responding incorrectly to more than one question potentially resulted in a $3 penalty for that session ($3 were subtracted from the final payment of $100); however, the participant who failed this check could read the bulletin once more and re-take the test to avoid losing the $3.

### 2.1.6. Baseline survey (at the end of the baseline period)

At the end of the baseline period, participants filled out an online survey through Qualtrics (Qualtrics, Provo, UT) measuring: A) 11 psychological factors (1. affective engagement, 2. collective efficacy, 3. perceived behavioural control, 4. social norm, 5. faith in institutions, 6. self-efficacy, 7. identity, 8. attitudes, 9. intentions, 10. openness/imagination, 11. theory of change), B) self-reported activism behaviour and C) demographics (figure 1*c*).

#### 2.1.6.1. 11 Psychological factors

For each factor, participants provided ratings on multiple items. Below is an example question for each of the 11 factors and their source. *Affective engagement*: 'When you think about a future impacted by climate change how strongly do you feel the following emotion? Fear…', rated from 1 (none at all) to 7 (very strongly) [12]. *Collective efficacy*: 'I feel confident about the capability of our society to address the climate crisis very well', rated from 1 (strongly disagree) to 7 (strongly agree) [16,28]. *Perceived behavioural control*: 'How much control do you have over whether you engage in climate activism?' from 1 (very little control) to 7 (a great deal of control) [17]. *Social norm*: 'If I engaged in environmental activism, people who are important to me would', from 1 (completely disapprove) to 7 (completely approve) [17]. *Faith in institutions*: 'How much do you trust the following group to do what is right in regard to the climate crisis? Governmental groups …', from 1 (never trust) to 7 (completely trust) [29]. *Self-efficacy*: 'I, personally, have the skills and resources to address climate change', from 1 (strongly disagree) to 7 (strongly agree) [30]. *Identity*: 'I see myself as a pro-environmentalist', from 1 (strongly disagree) to 7 (strongly agree) [31]. *Attitudes*: 'I think that engaging in climate activism is', from -3 (extremely bad) to +3 (extremely good)' [17]. *Intention*: 'I intend to engage in climate activism during the next 6 months', from 1 (extremely unlikely) to 7 (extremely likely) [17]. *Openness/Imagination*: 'I would like a job that requires following a routine rather than being creative', from 1 (disagree strongly) to 7 (agree strongly) [32]. *Theory of change*: 'If our society starts changing to help stop the climate crisis, will these changes start from politicians/governments or from people/grassroots movements?', from 1 (definitely politicians) to 7 (definitely people). This last factor is an *ad-hoc* measure of participants' belief on how successful change would be implemented: either as a bottom-up process stemming from grassroots action and expanding to the whole society or as a top-down process implemented by governments and authorities. We expected that belief in bottom-up change would more strongly increase the motivation to engage in action.

### 2.1.6.2. Baseline activism behaviours and other climate-related behaviours (self-report measures)

Participants reported: 1) their environmental education and leadership behaviours over the past six-week baseline period (e.g. 'I talk with others about climate change', rated from 1 (*not true at all*) to 4 (*completely true*) (modified from [33]), and 2) their emissions reduction behaviors (15 items), e.g. 'How often do you walk, bicycle, carpool or take public transportation instead of driving a vehicle by yourself?', rated from 1 (*never*) to 5 (*always*) [31].

### 2.1.6.3. Demographics

Participants reported their age, gender, education, income, ethnicity and political affiliation (see electronic supplementary materials for full measures).

### 2.1.7. Video intervention

Our video intervention was designed to boost the 11 psychological factors. The intervention consisted of 12 videos, each of which contained a mixture of two or more of the 11 psychological factors of interest. Each video was built on a theme (1. intro, 2. environmental threat, 3. human threat, 4. energy sources, 5. politics, 6. climate justice, 7. the climate movement, 8. victims and perpetrators, 9. neoliberalism and consumerism, 10. obstacles to engage, 11. how change happens, 12. imagine a climate-friendly world) and the 12 themes were selected based on the thematic curriculum of a college course on climate change taught by Dr Aron. The choice to build thematic videos, rather than building each video around one of the 11 factors, was done for two reasons: first, we were hoping to repeat the mobilizing effect of Dr Aron's class by adopting its thematic design which has repeatedly motivated students to climate action; and second, to prevent the participants from recognizing too easily in the videos the factors we were trying to boost.

We now give a flavour of how we tackled each psychological factor across the 12 videos: *affective engagement:* footage of the dramatic impacts of the climate crisis on people's health and living condition (e.g. communities devastated by floods); *collective efficacy*: footage of collective climate action such as the Sunrise movement campaigns that achieved a shift in political priorities; *social norms*: interviews with professors and peer-student activists talking about their experiences, bonds and mentorship within their activist community; *faith in institutions*: a combination of the history of unproductive international and national climate policy initiatives, combined with examples of current politicians that have an earnest climate crisis focus; *self-identity*: scenes that would engage, for example, self-identified naturalists and social justice advocates; *self-efficacy and attitudes*: testimonies of other students devoting themselves to making change; *theory of change*: interviews of activists and academics analysing the history of social movements and the feasibility of local climate activism in driving policy change; and *imagination*: hypothetical scenarios of a future world following a transition away from fossil fuels. The two other factors of *perceived behavioural control* and *intentions* were covered by including footage of student activists addressing the feasibility of climate activism as yet another extra-curricular activity.

During the intervention period, participants received two Qualtrics surveys per week for six weeks. Each Qualtrics survey began with a 20-minute video that they were required to watch from beginning to end (figure 1*d*). They were asked to avoid distractions and to put away their devices. The video software dictated the pace of viewing (i.e. they could not advance beyond the video without the total time of the video having passed). After the video, the participants were then prompted to read a 'climate events bulletin' which was identical to the climate event bulletins they received during the baseline period, but with new events (figure 1*d*). Again, at the top of the bulletin, participants were reminded that they were not penalized for not attending these events. They were then asked seven attention-check questions, five covering the video and two covering the event bulletin. Responding incorrectly to three or more questions resulted in a $3 penalty for that overall video session ($3 were subtracted from the final payment of $100), and the participants were redirected to watch the video again and to re-take the final test to avoid losing the $3 (they only had two attempts).

### 2.1.8. Follow-up survey

After the 12-week intervention, participants completed a follow-up survey very similar to the baseline survey but without demographics (figure 1*e*).

### 2.1.8.1. Activism behaviours

We embedded a study team member in the partner organizations. She monitored how many climate activism events were attended by each participant. The events were held on Zoom during this period due to the COVID-19 pandemic. She did so by comparing all sign-in names on Zoom calls with the names of the participants of our study, and if any match was found, one 'event score' was added for each participant attending. This was repeated at every event so that for all participants of our experiment, there was a record showing which events they attended. Participants also self-reported their environmental education and leadership behaviours and their emissions reduction behaviours.

### 2.1.9. Semi-structured interview

Following the confirmatory analyses, we conducted semi-structured interviews within two months from the end of the study (figure 1*f*) [34,35]. This was initially planned to gain a more nuanced understanding of how participants experienced the psychological factors that emerged as the best predictors of activism behaviour. Due to the lack of activism behaviour, these interviews were used to understand what held back the participants from engaging in the online events. The semi-structured interviews also explored participants' subjective beliefs about what caused any increase in the psychological factors from pre- to post-intervention. With these general aims in mind, and learning from each participant's quantitative results, we tailored an interview script. Possible questions could relate to changes or lack of change from baseline to follow-up, as well as the lack of engagement in climate action. An example script is shown in the electronic supplementary materials.

### 2.1.10. Exclusion criteria

The exclusion criteria were answering incorrectly to more than 10 attention-check questions (two bulletins per week, two questions per bulletin, for a total of 24 questions) during the baseline period or more than 24 attention check questions (two videos of five questions each +2 bulletins of two questions each per week, for six weeks, for a total of 84 questions). No participant was excluded for this reason.

### 2.1.11. Hypotheses

H1:  One or more of the 11 psychological factors related to climate activism would increase from baseline to follow-up.

We expected that factor 3 (faith in institutions) might go up or down after the intervention. For example, some participants may have concluded from the material we showed that policy elites are unreliable, given the general failure of UN and governmental policy over 30 years. Others might have felt more faith in institutions when they saw how the Sunrise movement boosted congressional action. We therefore maintained neutral expectations for how this factor may change and how this variation could affect participation.

H2:  Each behavioural outcome would increase from baseline to follow-up: (a) climate activism events attended (objective), (b) environmental education and leadership behaviours (self-report), and (c) emissions-reduction behaviours (self-report).

H3:  For each behavioural outcome, any change from baseline would be explained by changes in one or more of the psychological factors from the beginning to the end of the study.

### 2.1.12. Analytic plan

#### 2.1.12.1. Predictors

We calculated mean composites from the items within each of the 11 psychological factors. However, we note that factor 11 (theory of change) was novel, and Cronbach's $\alpha = 0.56$ for this factor, at both time points. To reach the minimum reliability score of $\alpha = 0.6$, one item was dropped, which led to $\alpha = 0.64$. All other composites were $\alpha > 0.6$ (see electronic supplementary materials for all alphas).

### 2.1.12.2. Outcomes

Objective participation in activism was computed for each participant as the mean number of events attended (from the events bulletin) during the baseline period and during the intervention period. The self-reported environmental education and leadership behaviours and emissions-reduction behaviours were computed separately for each timepoint as mean composites.

H1: We ran 11 paired-sample (pre-post) *t*-tests, one for each psychological factor composite, using Bonferroni-Holm for multiple comparisons correction.

H2: For each of the three behavioural outcomes we ran paired-samples *t*-tests between baseline and follow-up, using Bonferroni-Holm for multiple comparisons correction.

H3: Given the lack of change of the behavioural outcome from pre- to post-intervention, this analysis was dropped.

### 2.1.13. Exploratory analyses

#### 2.1.13.1. Baseline psychological factors

According to our pre-registered report, we would test whether the baseline psychological factors predicted the outcome behaviours. However, given the lack of behavioural outcome change from pre- to post-intervention, this analysis was also dropped.

#### 2.1.13.2. Semi-structured interview

We interviewed those participants who expressed their interest in this part of the study ($n = 40$). Interviews were conducted through online video via Zoom and lasted one hour. Audio of the interviews was recorded and transcribed for analysis, removing any identifiable information, and then destroyed. Interviews were coded and transcribed by NVivo Software. Following transcription, participants were contacted with a copy of the interview transcript for review and approval. We then conducted a content analysis of all transcripts, to identify common themes; we adhered to the guidelines of content analysis in [36]. Two coders started by familiarizing themselves with the data and collaborating over the generation of initial content codes. Following the establishment of the codebook, they each coded two specific transcripts (i.e. the same ones), discussed discrepancies and revised the codebook as necessary. The remaining transcripts were split between the two coders. Upon completion of the coding process, they then worked collaboratively to identify, review and define the themes that arose.

#### 2.1.13.3. *Post-hoc* follow-up survey

Five months after the end of the study, a follow-up survey was sent to all the participants, asking them to report whether they had participated in any event organized by a climate organization after the end of the study. If they answered yes, they were asked to list the name(s) of the organization(s) and the number of events attended. This follow-up survey was designed and delivered after stage 1 approval.

## 2.2. Results

We started by testing hypothesis **H1** that one or more of the 11 psychological factors related to climate activism would increase from baseline to follow-up. Three psychological factors significantly increased from pre- to post-intervention (after applying Bonferroni-Holm correction): affective engagement $t_{95} = 3.25$, $p = 0.02$, $d = 0.33$, collective efficacy $t_{95} = 4.94$, $p < 0.001$, $d = 0.50$ and self-efficacy $t_{95} = 4.73$, $p < 0.001$, $d = 0.48$ (figure 2). No other factors changed, all $ts(95) < 2.50$, all $ps > 0.17$ all $ds < 0.26$.

Pre-post change for all factors is shown in table 1.

For **H2**, our prediction was that each behavioural outcome would increase from baseline to follow-up: (a) climate activism events attended (objective), (b) environmental education and leadership behaviours (self-report) and (c) emissions-reduction behaviours (self-report). However, there was scarcely any change in behavioural outcomes. For objective attendance of activism events, only two participants joined during the intervention (one participant joined five events, and the other participant joined one event); self-reported environmental education and leadership had no change ($-0.02$ out of 7), and self-reported emissions reduction also had no change ($+0.01$ out of 7, all $ts$ (95) < 1.00, all $ps > 0.99$ all $ds < 0.10$).

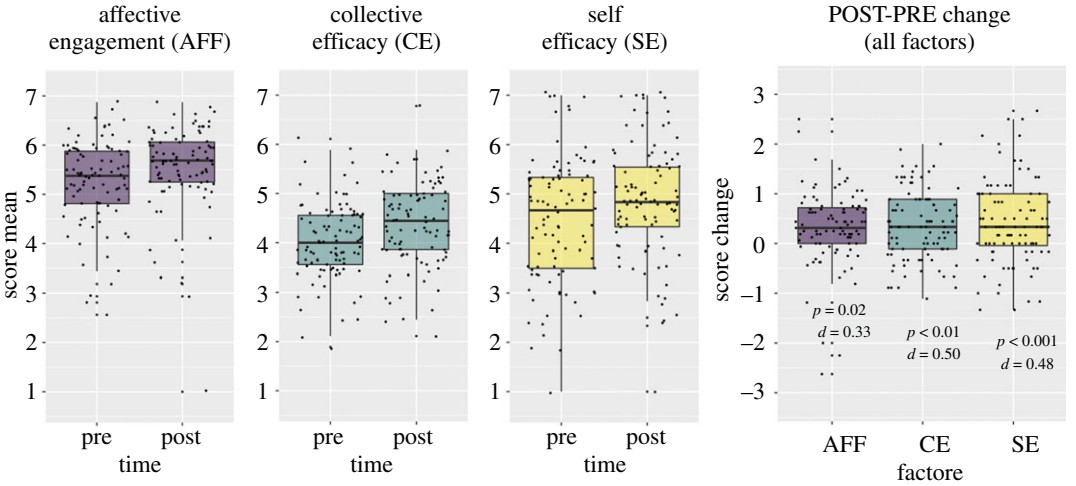

**Figure 2.** *Change in three psychological factors after the 12-video intervention. Note.* The boxes include the second and third quartiles divided by the median. The upper whiskers extend to the maximum value within 1.5 times the interquartile range over the 75th percentile, and the lower whiskers to the minimum within 1.5 times the interquartile range under the 25th percentile.

**Table 1.** Pre-post change for the 11 psychological factors and self-reported outcomes. Note. Significant *p*-values are in bold.

| factor | *M* pre | *s.d.* pre | *M* post | *s.d.* post | *t*(95) | *p* | *d* |
|---|---|---|---|---|---|---|---|
| attitudes | 5.74 | 1.21 | 5.98 | 1.05 | 1.89 | 0.67 | 0.19 |
| perceived behavioural control | 4.94 | 1.03 | 5.14 | 1.22 | 2.22 | 0.32 | 0.23 |
| social norm | 5.45 | 1.14 | 5.52 | 1.18 | 0.64 | 1 | 0.07 |
| faith in institutions | 3.63 | 0.89 | 3.48 | 0.88 | −1.98 | 0.56 | 0.20 |
| collective efficacy | 4.02 | 0.78 | 4.37 | 0.86 | 4.94 | **<0.01** | 0.50 |
| self-efficacy | 4.48 | 1.28 | 4.89 | 1.16 | 4.74 | **<0.01** | 0.48 |
| affective engagement | 5.25 | 0.86 | 5.53 | 0.89 | 3.25 | **0.02** | 0.33 |
| identity | 4.13 | 0.62 | 4.23 | 0.78 | 1.82 | 0.79 | 0.19 |
| intentions | 4.24 | 1.61 | 4.36 | 1.71 | 0.92 | 1 | 0.09 |
| openness | 4.33 | 0.81 | 4.44 | 0.86 | 2.44 | 0.18 | 0.25 |
| theory of change | 5.25 | 1.10 | 5.52 | 1.07 | 2.33 | 0.24 | 0.24 |
| outcome: emission red. | 3.40 | 0.43 | 3.42 | 0.53 | 0.28 | 1 | 0.02 |
| outcome: edu./leadership | 1.90 | 0.54 | 1.89 | 0.55 | −0.50 | 1 | 0.05 |

### 2.2.1. Final interviews

Forty participants agreed to be interviewed at the end of the study. Based on the quantitative results above, the final interview script was structured to address two main questions: (1) what prevented participants from attending the online activism events? and (2) what led to the change in the psychological factors that a) increased on average across participants (affective engagement, self-efficacy and collective efficacy) and b) increased for an individual participant (i.e. at least +2 out of 7 on a factor)? Finally, participants were asked their opinion on ways to better design this study to achieve greater activist engagement.

#### 2.2.1.1. Interviewee responses on the lack of behavioural change

Interviewees were told that most of the participants in the study did not attend the activism events advertised during the study time. They were asked why they thought most people did not attend and

**Table 2.** Follow-up interviews ($n = 40$). *Note.* Answers given by three or fewer participants are not shown.

| perceived obstacles | description | N | % |
|---|---|---|---|
| Zoom fatigue | tired of online sessions such as classes during the pandemic | 30 | 75 |
| lack of normal socialization due to Zoom | events lacked the social component typically found within organizing/campus activities | 25 | 62 |
| workload | school workload | 23 | 57 |
| other responsibilities | responsibilities aside from schoolwork (i.e. a job, babysitting, other community orgs) | 16 | 40 |
| not enough bandwidth or energy | feels too drained of energy to engage with climate activism | 13 | 32 |
| didn't know anyone | doesn't know people going to the event | 11 | 27 |
| events not required or emphasized | the event was seen as voluntary and not necessary to complete the study | 9 | 22 |
| uninteresting events or advertisement | the events looked generic each week and overall uninteresting | 9 | 22 |
| laziness or lack of care | does not care or is too lazy to attend | 8 | 20 |
| didn't feel like they could contribute | doesn't feel like they have anything to individually offer climate orgs/activism (reflecting individual efficacy) | 6 | 15 |
| imposter syndrome | feels like an imposter among people who are more knowledgeable or passionate about climate activism | 5 | 12 |
| not sure they could make a difference | isn't sure that their efforts within an event will make a difference (reflecting more collective efficacy) | 4 | 10 |
| time difference | couldn't attend the events due to time differences (if abroad) | 4 | 10 |

why they themselves did not feel compelled to attend. As personal reasons were usually merged with speculations about other people's reasons, both these answers are compiled together in table 2. The most common obstacles were (1) Zoom fatigue, owing to the COVID-19 pandemic (during this study, students had been attending classes on Zoom for over a year); (2) perception of a lack of opportunities to socialize normally given the climate events were only on Zoom; (3) school workload and (4) being too busy due to other responsibilities such as a job or community organizing.

### 2.2.1.2. Pre-post psychological factor changes

The participants were asked why they thought affective engagement, collective efficacy and self-efficacy changed at the group level. A description of these results can be found in the electronic supplementary material (table S1), along with a description of why they thought some psychological factors increased for them personally (electronic supplementary material, table S2).

## 2.3. *Post-hoc* and exploratory analyses

### 2.3.1. Follow-up survey

After five months, only 7% (three) of the 40 participants reported having attended between one and five activism events organized by the UCSD Green New Deal (climate organization), and one participant said they attended three events at the not-monitored organization Grove. Based on the attendance records of UCSD Green New Deal, we confirmed that one participant became a sustained member of the organization. It was not possible to verify attendance for the other two participants. However, not all the organization's events happening over those five months required sign-up sheets, so it is unknown whether they participated in one of the unmonitored events. Regardless, event attendance was nearly zero after the study.

### 2.3.2. Risk perception

Affective engagement had two subcategories: risk perception (questions such as 'How likely do you think it is, from 1 to 7, that a) worldwide, many people's standard of living will decrease, b) worldwide water shortages will occur….') and emotional response (questions such as 'When you think about a future impacted by climate change how much, from 1 to 7, of the following emotions do you feel? Guilt, Sadness, Fear, Shame…'). In the main analysis, risk perception and emotional response were collapsed into one composite score for affective engagement. In this *post-hoc* phase these items were analysed separately, which revealed that while risk perception changed from pre- to post-intervention ($p < 0.001$, $d = 0.43$), the emotional response did not ($p = 1$, $d = 0.1$). Additionally, the pre-post change for risk perception was greater than the pre-post change for emotional response ($t_{95} = 3.00$, $p = 0.003$, $d = 0.3$).

# 3. Study 1 discussion

This study tested whether a longitudinal video intervention would trigger non-activists to join online climate action events and to test which psychological factors might account for such behavioural change. After the six-week video intervention, three psychological factors increased: affective engagement, collective efficacy, and self-efficacy. Exit interviews of 40 participants suggested that the main trigger of these changes was the video content, in particular the information about extreme weather events, health impacts and collective and individual activism.

Contrary to our expectations, only two participants joined the activist events (all of which were online given the COVID-19 pandemic). After five months, one participant had become a sustained activist in a local organization. The interviews suggested that poor participation was overwhelmingly due to Zoom fatigue and the participants' perception that the Zoom format precluded typical social interactions. This opened the question of whether the intervention was weak due to this lack of social interaction or to the absence of change in the other eight psychological factors. Therefore, after the Registered Report Stage 1 In-Principle Acceptance and with the editor's permission, we ran an exploratory follow-up study with an in-person intervention where participants watched the videos in a social setting and could participate in on-campus activism events.

## 3.1. Unregistered pilot study 2

This was an in-person, six-week replication of Study 1 and did not include a baseline. The aim was to test whether the in-person format triggered any activist participation.

### 3.1.1. Methods

We recruited 66 participants assuming a similar dropout rate to Study 1 (about 30% or greater, given the more effortful in-person tasks), aiming for an $N = 40$ final sample. This small sample (smaller than the $N$ prescribed by our power analysis for Study 1) seemed appropriate for a small pilot study with no planned comparison between two groups (experimental and baseline). 38 completed the full study (a 42% dropout rate). Recruitment was carried out as in Study 1, but this time the final reward amount was $150 per person to facilitate recruitment under pandemic conditions.

As in Study 1, participants first underwent a screening survey to exclude those not believing in anthropogenic global heating and those who attended more than one climate activism event prior to the study (figure 1). Those passing the screening signed a consent form agreeing to complete online surveys and to attend a 1.5 h in-person event every week for six weeks. Everyone first completed a baseline survey measuring the same 11 psychological factors and self-reported climate-related behaviors as Study 1. Then, every Monday night, participants gathered in a classroom at UC San Diego. Here, a study member coordinated an ice-breaking activity: solving riddles in groups of three or four. Then, she showed two 25-min videos (the same videos from Study 1), after which the participants took a four-question comprehension quiz on their mobile phone. They were prompted to respond to the last two questions of the quiz collectively in groups of three to four to engage each other in a conversation about the video content. After this, an activist guest from the climate org UCSD Green New Deal came into the room and described a bulletin of events for her organization for the following five days. A comprehension quiz followed with participants responding to the last two questions in groups. If participants did not show up at the Monday session or did not complete any of the comprehension quizzes, they incurred a $20 penalty. At the activism events, a confederate

tracked attendance by having all attendees fill in a sign-up sheet. At the end of the six weeks, participants again completed the survey measuring the 11 psychological factors and self-reported behaviors. There was a group debrief activity during the last in-person session.

We hypothesized that the in-person intervention would increase some of the 11 psychological factors and attendance of activist events.

### 3.1.2. Results

Two psychological factors increased after the intervention (after Bonferroni-Holm correction): self-efficacy $t_{37} = 3.62$, $p = 0.01$, $d = 0.33$ and identity $t_{37} = 3.60$, $p < 0.001$, $d = 0.59$. None of the other factors changed (all $ts_{37} < 2.97$, all $ps > 0.05$, all $ds < 0.49$).

Only two participants (5%) engaged in objectively verified activist events (they attended one event each). During the final group debrief, participants most often reported that they did not participate because of the lack of time, scheduling conflicts and lack of payment. Self-reported activist behaviour did not increase, neither for environmental education and leadership (+0.09 out of 7), nor for emissions reduction behaviors (+0.21 out of 7) (all $ts_{37} < 1.60$, all $ps > 0.10$ all $ds < 0.30$).

### 3.1.3. Discussion

Study 2 was an additional, in-person intervention study where videos were watched in a social context, event bulletins were presented by a real activist (rather than emailed in a written list), and activism events were held in person on campus rather than on Zoom. However, only two participants (5%) each joined a single event. This small study suggests that including a social component, in-person study events, and activist events is not sufficient to trigger attendance at activist events, at least with this study design in this population and during the late-2021 pandemic.

## 4. General discussion

Three psychological factors were boosted by the Study 1 intervention (self-efficacy, collective efficacy and affective engagement) and only two of 96 participants attended activist events (2%). The main reasons they gave for not attending were Zoom fatigue and a perceived lack of social interaction at online events. The in-person intervention in Study 2 increased two factors: self-efficacy (like in Study 1) and identity, and only two out of 38 participants attended an activist event (5%).

### 4.1. Change in psychological factors

Across both studies, the video intervention boosted self-efficacy, such that participants felt empowered and more aware that their skills could contribute to climate activism. The intervention also boosted affective engagement, collective efficacy (Study 1), and identity (Study 2). Only four of the 11 factors changed in either study, suggesting that changes this size in these four factors alone were not sufficient to trigger action. Re-designing the videos might help to boost all 11 factors. Future research could employ a professional videographer, create content that is more local to the participants and use forms of narrative structure, perhaps by involving actors. Or, additional factors might need to be discovered and targeted. It remains possible that large increases in the 11 factors might not be sufficient to trigger action when participants do not have the time or financial resources to volunteer for activism.

### 4.2. Recruitment incentives

When advertising the studies (with a reward of $100 for Study 1 and $150 for Study 2) we hid the climate crisis content to attract a neutral (and generalizable) student audience rather than a self-selecting audience already interested in climate action. For Study 2 the payment was increased to $150 because few students were signing up, probably due to pandemic conditions. The informal debriefing for Study 2 strongly indicated that participants were short of money and time and expected payment to join the activist events.

In our assessment, the central reason why our two studies were not effective in driving more people to participate in climate activism was that they were participating mostly for payment. This reflects the socioeconomic status and material realities of a segment of the UC San Diego undergraduate population,

many of whom both attend school and work jobs. Students did appear to care about the climate, but they appeared not to have the time and resources to enter these particular activist spaces. This observation leads to the fundamental question of how to overcome the near-universal barriers of time and financial resources to engage in activism.

One potential incentive to join action for those lacking time or financial resources could be the support system that is often created inside an activist group—something our study did not encourage. Our personal observations point to emotional and social benefits of participation. For example, when activists share values and struggles inside the activist space, they become empathetic with each other and can learn solidarity skills of mutual aid and support to overcome engagement obstacles. For example, they can cover each other's work shifts to allow one to go to a rally or help each other find jobs within activism or advocacy. These connections with other activists also lead to valuable professional networks. Accordingly, interventions might trigger more engagement by advertising these benefits or designing the intervention to foster these exchanges. We also recommend providing more event times that fit busy schedules, and specifically recruiting participants already interested or engaged in climate activism.

# 5. Conclusion

The intensive video intervention increased several psychological factors associated with collective action on climate, but only a few participants attended activist events. The main weakness in this design may have been the reliance on paid study participation: we appear to have inadvertently selected some of the financially challenged students in our community who had little time and resources for volunteer activities. The study design could be strengthened in multiple ways from better video-making to creating different participation opportunities.

The central question of this study remains highly relevant: how can people be moved to collective action to protect our climate during a climate collapse? This study sheds light on how difficult it is to trigger this behaviour, which is even more reason for psychology studies using longitudinal designs and objective measurements of key behaviors.

Ethics. Our procedures were approved by the University of California, San Diego, Human Research Protections Programme, with project no. IRB #201545.

Data accessibility. All data and code are publicly available at https://osf.io/38vkz/?view_only=fbc1a81292a5404 c9418c83c5fa06c93. **Stage 1 Registered Report:** doi:10.31234/osf.io/jqz29.

Electronic supplementary material is available online [37].

Authors' contributions. A.C.: conceptualization, data curation, formal analysis, funding acquisition, investigation, methodology, project administration, resources, software, supervision, validation, visualization, writing—original draft, writing—review and editing; C.B.: methodology, writing—review and editing; S.H.: data curation, formal analysis, investigation, methodology, software, visualization; E.M.: data curation; A.A.: conceptualization, funding acquisition, methodology, resources, supervision, writing—review and editing.

All authors gave final approval for publication and agreed to be held accountable for the work performed therein.

Conflict of interest declaration. The authors declared no potential competing interests with respect to the authorship and/or publication of this article.

Funding. The research was supported by the Yankelovich Center for Social Science Research at the University of California, San Diego.

Acknowledgements. We thank Debra Lindsay for helping us identify the rating scales to measure the 11 factors of interest, and for her technical support building some of the surveys.

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
