## [Peer Review File · Royal Society Open Science]

Review History

RSOS-210006.R0 (Original submission)

Review form: Reviewer 1

Do you have any ethical concerns with this paper?

Yes

Recommendation?

Accept with minor revision

Comments to the Author(s)

Congratulations on studying the mechanisms of activism. Activism is an often-overlooked category of pro-environmental behavior, with huge potential to lead to large scale system change. I recommend minor revisions such that there are clearer definitions of key terms, and more appropriate quality assurance techniques for the qualitative component (see Appendix A).

Review form: Reviewer 2

Do you have any ethical concerns with this paper?

No

Recommendation?

Major revision

Comments to the Author(s)

1. The scientific validity of the research question(s)

I think the research question is well-founded in existing scientific research. The psychological factors that are chosen are frequently reported to play a role in predicting climate activism, while interesting relatively new ones (e.g. Theory of change) are included, too. Additionally, I think the research question has important practical implications. From what I understood, the idea was to find out which of the 11 psychological factors are more and which are less important in causing environmental activism. I think answering this question can be very important in applied climate activism, in order to know how to best use limited resource for mobilization. Furthermore, I think a video intervention is especially practical regarding the low costs, but also regarding the high feasibility during the covid-19 pandemic. A particular strength of the article is the aim to measure climate activism through a behavioral, "objective" measure, instead of relying on self-reported climate activism. Moreover, including environmental education and leadership behaviors as outcome variables could provide interesting new nuances. Including semi-structured interviews could furthermore be a very meaningful measure to find out the strengths and limitation of the intervention.

2. The logic, rationale, and plausibility of the proposed hypotheses

H1: "Some of the 11 psychological factors related to climate activism will increase from baseline to follow-up."

While this is a valid research question, this does not constitute a testable or falsifiable hypothesis. This hypothesis would be confirmed as soon as one factor increases (non-significantly) from baseline to follow-up, even if all other factors don't increase or even decrease. The hypothesis should be made more specific in order to prevent undisclosed flexibility in the interpretation of results. With the current wording, what constitutes a confirmation of the hypothesis is greatly open to interpretation by the researchers. The same applies to H3.

Logic:

- Is the factor Faith in Institutions expected to increase, too? The video intervention could lead to an increase and to a decrease of faith in institutions (intervention: "a combination of the history of unproductive international and national climate policy initiatives, combined with examples of current politicians that have an earnest climate crisis focus").

- What does it mean if the factor "theory of change" increases? Politicians or people?
Could be more specific.

H2: "Each behavioral outcome will increase from baseline to follow-up: a) climate activism events attended (objective, self-reported)"

For objective climate activism events attended, from my understanding, there is no baseline measurement, thus, it can't increase from baseline to follow-up. Otherwise, the objective climate activism measure is not included in any hypothesis, while (judging from the introduction) it is supposed to be a central point of the current study. What are the plans for interpreting and analyzing this measure? I think, trying to measure climate activism more objectively is a great idea. However, I was wondering what the plan for interpretation is, if self-report and objective

measures differ? What would speak more towards a confirmation of the hypothesis? What will be regarded as the more valid measure to draw conclusions from? In order to avoid being able to change what constitutes a confirmation of the hypothesis after having analyzed the results, it should be specified in advance, which measure is regarded as the most important outcome measure.

3. The soundness and feasibility of the methodology and analysis pipeline (including statistical power analysis where applicable)

No pre-/ baseline measure for “objective” activism behavior:

In the introduction, it is clearly noted that “noise and bias in self-report measures of activism behavior” exists and it is important to assess activism behavior objectively, instead of relying on self-report. I really like that idea and think it is a very interesting approach to look at actual participation in activism events. However, it strikes me as unusual then, that the study protocol does not include (or at least does not report) a baseline measure of “objective” activism behavior. If the study protocol does include a baseline measure of objective activism behavior, I strongly encourage to make this more explicit in the methods. If no baseline measurement of objective activism behavior is included, this strongly limits the interpretation with regards to the effectiveness of the intervention. Including a baseline measurement could be done by including a (equally long) baseline-phase, where participants receive no video intervention or booklet, (or only receive the booklet about activism events that happen this week,) but “objective” attendance to activism events is still recorded. However, not including such a baseline measure does not allow for causal inferences about objective activism behavior. It is a great goal to measure activism behavior “objectively”, but if there is no data on objective activism behavior available from before the intervention, the analyses for the effectiveness of the intervention will have to rely again on self-reports of activism behavior.

Confounding variable systematically varying with the intervention: climate events bulletin?

I was wondering whether the climate events bulletin was supposed to be part of the intervention, or whether the video intervention was assumed to work independently of the climate events bulletin? If so, a baseline phase for the bulletin only (without the video intervention) would be desirable, in order to be able to distinguish the effect of the event bulletin from the effect of the video intervention. It could be possible that all observed changes in psychological factors and behavioral outcomes are solely the result of reading the climate events bulletin. For example, making it salient through the bulletin that other people participate in climate activism could influence social norms, as well as the behavioral outcomes. With the current experimental setup, there is no way to rule this confounding variable out. I do understand that it is important to provide participants with enough opportunities to actually carry out environmental activism behavior, and to guide them towards the climate groups that are working together with the researchers. However, without any control/ baseline phase which only includes the bulletin, there is no way to draw a clear causal interpretation of the results to the video intervention.

Validation of the intervention:

The current design might not allow to clearly identify all of the tested factors that may cause climate activism. Some might be not identified (or misidentified as being not important) due to the intervention not being sufficiently pilot-tested and validated.

It would make sense to first validate the video intervention and see whether all factors are influenced equally over the course of the 12 weeks. Then – after making sure that the intervention influences each factor comparably effectively – in a separate sample, information about activism events (bulletin) could be given and activism behavior could be assessed.

Without this validation, it is difficult to interpret the relative importance of the factors. For example, an increase in affective engagement could be found to predict an increase in activism

behavior much better than all the remaining 10 factors. However, it is difficult to say whether this would be due to the intervention being particularly effective at manipulating affective engagement (and not really influencing the other 10 factors), or whether affective engagement actually is the most important predictor for activism. This concern should be addressed/considered, and it should be communicated how the authors plan to interpret those results.

The current design can only provide insights on how effective the intervention was to manipulate the 11 factors and whether factors that have been effectively manipulated predict activism behavior.

4. Would the clarity and degree of methodological detail be sufficient to replicate exactly the proposed experimental procedures and analysis pipeline?

In order to fully understand the intervention (and replicate the study), more information about the content of the other 11 videos would be needed. For example, do the participants watch the same video every week? Was the first video (the example video) a general introductory video and the following 11 weeks will target each of the 11 identified factors separately? If so, was there a specific order chosen, based on ideas about how the factors might influence each other? Would the intervention work in a different order, too?

Furthermore, more information about how the cooperating climate groups are collecting data about the attendance of participants would be desirable for replication. Additionally, in order to prevent potential privacy concerns about data collection (since the study did not seem to be reviewed by an ethics committee), more transparency would be great.

5. Do the authors provide a sufficiently clear and detailed description of the methods to prevent undisclosed flexibility in the experimental procedures or analysis pipeline?

- The questions for the semistructured interviews seem to be adapted depending on the results of the quantitative data. If so, it would be good to have a little more transparency as to what questions are chosen and why/ depending on what results. Additionally, it would be great to be provided with the reasons for why only people who attended at least one climate activism event are considered for the interviews. This is a little confusing since an interest in investigating perceived barriers is expressed several times throughout the article.

- "If any composite score within each timepoint has a Cronbach's alpha \leq .5, questions will be removed until the alpha is above this threshold. If there is no such solution, a single face-valid item will be chosen prior to hypothesis testing." From my understanding, for at least some of the factors, previously validated scales are used. Removing questions to increase Cronbach's alpha would undermine the practice of validating questionnaires (and therefore reduce the scale's validity), especially if this results in only one face valid item left to be used.

- Analytic plan H2: "If the subjective and objective event attendance is correlated at $r \geq .7$, the test with the subjective outcome will be dropped to reduce the number of tests." Does this mean, the plan is to run a paired sample t-test between baseline subjective and follow-up objective activism? (Since there is no measure of baseline objective activism?)

6. Whether the authors have considered sufficient outcome-neutral conditions (e.g. positive controls) for ensuring that the results obtained are able to test the stated hypotheses

Not really applicable here?

Review form: Reviewer 3

Do you have any ethical concerns with this paper?

No

Recommendation?

Major revision

Comments to the Author(s)

This pre-registered study proposes to identify the psychological factors that are most strongly associated with increased engagement in real-world collective climate action as a result of a 12-week intervention. Rather than testing a specific theory about the link between psychological factors and action, the authors propose two novel methodological tools: a video that attempts to boost 11 identified psychological factors and a behavior-tracking methodology to measure real-world collective action. The proposed single-cohort pre-post longitudinal study will recruit 160 participants aged 18-30 who will complete an online survey assessing the 11 psychological factors of interest, as well as self-reporting baseline activism behaviors and demographics. They will complete a 12-week video-based intervention, during which their activism will be monitored. At the end of the 12 weeks, they will complete the questionnaires for a second time. Proposed analyses will assess the pre-post change in the psychological factors, in behavioral outcomes (objectively monitored and self-reported), and the relationship between these changes.

The research question is timely and important and is well-motivated by the introduction. The methodology does a good job of outlining the design, the sample size, procedures related to outliers, and proposed analyses, including exploratory analyses - though some key gaps, including specification of DVs, need to be addressed, as outlined below. The issue of power is addressed. The proposed methodology and analysis pipeline appear largely sound and feasible, and are able to test the stated hypotheses, though I have some specific questions and concerns, which are outlined below. If these concerns are addressed, the project is likely to generate a rich and important dataset that will demonstrate whether video-based interventions can boost collective climate action, and will give new insights into the psychological factors that contribute to such change.

1. In the abstract and introduction, the authors highlight a lack of causal evidence regarding the psychological factors that predict collective climate action. A key drawback of the proposed single-cohort pre-post design, however, is that the absence of a control group will preclude causal inferences. While I sympathize with the arguments made regarding the sample size and cost entailed by the inclusion of a control group, the fact that the lack of a control group weakens claims to causal links should be acknowledged and, as a result the authors should perhaps rethink their focus on causal evidence.
2. The 11 psychological factors that will be investigated are not well motivated in the introduction - there is general reference to psychological factors rather than specific reference to the 11 that will be examined here. The rationale for these specific 11 factors should be expanded.
3. Further, it is unclear how the questionnaire that assesses these 11 factors was developed - it appears that questions were selected from previous studies and instruments, but how this was done, and whether any effort has been made to pilot this instrument is not reported. There is very unequal treatment of the 11 factors in the questionnaire - each factor is assessed with a different number of items (e.g., some factors are assessed with three questions, some with as many as 16) and using different types of response scales (some -3 to 3; some 1 to 7). The variable psychometric characteristics could pose a significant confound for any attempt to identify the factors that contribute to action. It is not clear why there are such differences, what their potential impact is, and how they will be controlled for or any potential impact mitigated against - other than the proposal to exclude questions that reduce Cronbach's alpha or to select "a single face-valid item".

4. Further details on the video intervention are required. First - as is the case for the questionnaire, it is unclear how the video intervention was developed and whether any attempt has been made to validate it - for example to assess the extent to which the 11 psychological factors are present or evoked by the video. It is not clear whether the participants will see the same or a different video each week. If the video is different, how will the quality/extent to which these factors are targeted be controlled across weeks? Will order of presentation be counterbalanced across participants?
5. Another shortcoming of existing research highlighted by the authors is the reliance on self-report or intentions rather than objectively measured participation - the authors proposed a novel "behavior-tracking methodology to measure real-world participation in climate action." This methodology is scantily described, however. It is not clear how the "partner organizations will monitor how many climate activism events are attended by each participant." This seems practically and ethically challenging to do and further details are needed. It seems likely that the authors will ultimately have to rely on self-reported participation, which again weakens the claims regarding novelty and advance on previous work. If the behavior-tracking methodology is successful, what will be measured, over what time scale, and how this will be included as an outcome variable in the analyses is not specified.
6. In the section on page 11 addressing outcome variables, climate activism events attended are not included as an outcome variable. As mentioned in the previous comment, it is not clear how the objective measure will be computed.
7. How will demand characteristics be addressed?
8. The study has the potential to generate a very rich and valuable dataset - a further potential exploratory analysis might include factor analyses of the questionnaire data, which might aid in data reduction, and in the identification of the most important factors/families of factors.

Decision letter (RSOS-210006.R0)

Dear Dr Castiglione,

The Editors assigned to your Stage 1 Registered Report ("Discovering the psychological building blocks underlying climate action - a longitudinal study of real-world activism") have now received comments from reviewers. We would like you to revise your paper in accordance with the referee and editors suggestions which can be found below (not including confidential reports to the Editor). Please note this decision does not guarantee eventual acceptance.

Please submit a copy of your revised paper within three weeks (i.e. by the 18-Mar-2021). If deemed necessary by the Editors, your manuscript will be sent back to one or more of the original reviewers for assessment.

When submitting your revised manuscript, you must respond to the comments made by the referees and upload a file "Response to Referees" in "Section 2 - File Upload". Please use this to

document how you have responded to the comments, and the adjustments you have made. In order to expedite the processing of the revised manuscript, please be as specific as possible in your response.

Kind regards,
Professor Chris Chambers
Royal Society Open Science
openscience@royalsociety.org

Associate Editor Comments to Author (Professor Chris Chambers):

Associate Editor: 1

Comments to the Author:

Thank you for your patience during this challenging time for reviewers. Three expert reviewers have now assessed the manuscript, and while they are unanimous in praising the value and importance of the research question, their assessments are also very critical of the specific rationale and methodology proposed. Headline issues include conceptual clarity and scope, precision of the hypotheses, justification (and lack of validation) of the specific components of the intervention, degree of methodological detail provided, and rigour of experimental control (including lack of a control group and consequent limitations for drawing causal conclusions). Overall, I believe that these issues can at least in principle be addressed, and Stage 1 IPA is achievable, although there is a some significant work to be done. I would therefore like to offer the authors the opportunity of submitting a Major Revision.

Comments to Author:

Reviewer: 1

Comments to the Author(s)

Congratulations on studying the mechanisms of activism. Activism is an often-overlooked category of pro-environmental behavior, with huge potential to lead to large scale system change. I recommend minor revisions such that there are clearer definitions of key terms, and more appropriate quality assurance techniques for the qualitative component.

(Detailed comments from Reviewer 1 are attached)

Reviewer: 2

Comments to the Author(s)

1. The scientific validity of the research question(s)

I think the research question is well-founded in existing scientific research. The psychological factors that are chosen are frequently reported to play a role in predicting climate activism, while interesting relatively new ones (e.g. Theory of change) are included, too. Additionally, I think the research question has important practical implications. From what I understood, the idea was to find out which of the 11 psychological factors are more and which are less important in causing environmental activism. I think answering this question can be very important in applied climate activism, in order to know how to best use limited resource for mobilization. Furthermore, I think a video intervention is especially practical regarding the low costs, but also regarding the high feasibility during the covid-19 pandemic. A particular strength of the article is the aim to measure

climate activism through a behavioral, “objective” measure, instead of relying on self-reported climate activism. Moreover, including environmental education and leadership behaviors as outcome variables could provide interesting new nuances. Including semi-structured interviews could furthermore be a very meaningful measure to find out the strengths and limitation of the intervention.

2. The logic, rationale, and plausibility of the proposed hypotheses

H1: “Some of the 11 psychological factors related to climate activism will increase from baseline to follow-up.”

While this is a valid research question, this does not constitute a testable or falsifiable hypothesis. This hypothesis would be confirmed as soon as one factor increases (non-significantly) from baseline to follow-up, even if all other factors don’t increase or even decrease. The hypothesis should be made more specific in order to prevent undisclosed flexibility in the interpretation of results. With the current wording, what constitutes a confirmation of the hypothesis is greatly open to interpretation by the researchers. The same applies to H3.

Logic:

- Is the factor Faith in Institutions expected to increase, too? The video intervention could lead to an increase and to a decrease of faith in institutions (intervention: “a combination of the history of unproductive international and national climate policy initiatives, combined with examples of current politicians that have an earnest climate crisis focus”).
- What does it mean if the factor “theory of change” increases? Politicians or people? Could be more specific.

H2: “Each behavioral outcome will increase from baseline to follow-up: a) climate activism events attended (objective, self-reported)”

For objective climate activism events attended, from my understanding, there is no baseline measurement, thus, it can’t increase from baseline to follow-up. Otherwise, the objective climate activism measure is not included in any hypothesis, while (judging from the introduction) it is supposed to be a central point of the current study. What are the plans for interpreting and analyzing this measure? I think, trying to measure climate activism more objectively is a great idea. However, I was wondering what the plan for interpretation is, if self-report and objective measures differ? What would speak more towards a confirmation of the hypothesis? What will be regarded as the more valid measure to draw conclusions from? In order to avoid being able to change what constitutes a confirmation of the hypothesis after having analyzed the results, it should be specified in advance, which measure is regarded as the most important outcome measure.

3. The soundness and feasibility of the methodology and analysis pipeline (including statistical power analysis where applicable)

No pre-/ baseline measure for “objective” activism behavior:

In the introduction, it is clearly noted that “noise and bias in self-report measures of activism behavior” exists and it is important to assess activism behavior objectively, instead of relying on self-report. I really like that idea and think it is a very interesting approach to look at actual participation in activism events. However, it strikes me as unusual then, that the study protocol does not include (or at least does not report) a baseline measure of “objective” activism behavior. If the study protocol does include a baseline measure of objective activism behavior, I strongly encourage to make this more explicit in the methods. If no baseline measurement of objective activism behavior is included, this strongly limits the interpretation with regards to the effectiveness of the intervention. Including a baseline measurement could be done by including a (equally long) baseline-phase, where participants receive no video intervention or booklet, (or

only receive the booklet about activism events that happen this week,) but “objective” attendance to activism events is still recorded. However, not including such a baseline measure does not allow for causal inferences about objective activism behavior. It is a great goal to measure activism behavior “objectively”, but if there is no data on objective activism behavior available from before the intervention, the analyses for the effectiveness of the intervention will have to rely again on self-reports of activism behavior.

Confounding variable systematically varying with the intervention: climate events bulletin?

I was wondering whether the climate events bulletin was supposed to be part of the intervention, or whether the video intervention was assumed to work independently of the climate events bulletin? If so, a baseline phase for the bulletin only (without the video intervention) would be desirable, in order to be able to distinguish the effect of the event bulletin from the effect of the video intervention. It could be possible that all observed changes in psychological factors and behavioral outcomes are solely the result of reading the climate events bulletin. For example, making it salient through the bulletin that other people participate in climate activism could influence social norms, as well as the behavioral outcomes. With the current experimental setup, there is no way to rule this confounding variable out. I do understand that it is important to provide participants with enough opportunities to actually carry out environmental activism behavior, and to guide them towards the climate groups that are working together with the researchers. However, without any control/ baseline phase which only includes the bulletin, there is no way to draw a clear causal interpretation of the results to the video intervention.

Validation of the intervention:

The current design might not allow to clearly identify all of the tested factors that may cause climate activism. Some might be not identified (or misidentified as being not important) due to the intervention not being sufficiently pilot-tested and validated.

It would make sense to first validate the video intervention and see whether all factors are influenced equally over the course of the 12 weeks. Then – after making sure that the intervention influences each factor comparably effectively – in a separate sample, information about activism events (bulletin) could be given and activism behavior could be assessed.

Without this validation, it is difficult to interpret the relative importance of the factors. For example, an increase in affective engagement could be found to predict an increase in activism behavior much better than all the remaining 10 factors. However, it is difficult to say whether this would be due to the intervention being particularly effective at manipulating affective engagement (and not really influencing the other 10 factors), or whether affective engagement actually is the most important predictor for activism. This concern should be addressed/ considered, and it should be communicated how the authors plan to interpret those results.

The current design can only provide insights on how effective the intervention was to manipulate the 11 factors and whether factors that have been effectively manipulated predict activism behavior.

4. Would the clarity and degree of methodological detail be sufficient to replicate exactly the proposed experimental procedures and analysis pipeline?

In order to fully understand the intervention (and replicate the study), more information about the content of the other 11 videos would be needed. For example, do the participants watch the same video every week? Was the first video (the example video) a general introductory video and the following 11 weeks will target each of the 11 identified factors separately? If so, was there a specific order chosen, based on ideas about how the factors might influence each other? Would the intervention work in a different order, too?

Furthermore, more information about how the cooperating climate groups are collecting data about the attendance of participants would be desirable for replication. Additionally, in order to prevent potential privacy concerns about data collection (since the study did not seem to be reviewed by an ethics committee), more transparency would be great.

5. Do the authors provide a sufficiently clear and detailed description of the methods to prevent undisclosed flexibility in the experimental procedures or analysis pipeline?

- The questions for the semistructured interviews seem to be adapted depending on the results of the quantitative data. If so, it would be good to have a little more transparency as to what questions are chosen and why/ depending on what results. Additionally, it would be great to be provided with the reasons for why only people who attended at least one climate activism event are considered for the interviews. This is a little confusing since an interest in investigating perceived barriers is expressed several times throughout the article.

- "If any composite score within each timepoint has a Cronbach's alpha $< .5$, questions will be removed until the alpha is above this threshold. If there is no such solution, a single face-valid item will be chosen prior to hypothesis testing." From my understanding, for at least some of the factors, previously validated scales are used. Removing questions to increase Cronbach's alpha would undermine the practice of validating questionnaires (and therefore reduce the scale's validity), especially if this results in only one face valid item left to be used.

- Analytic plan H2: "If the subjective and objective event attendance is correlated at $r > .7$, the test with the subjective outcome will be dropped to reduce the number of tests." Does this mean, the plan is to run a paired sample t-test between baseline subjective and follow-up objective activism? (Since there is no measure of baseline objective activism?)

6. Whether the authors have considered sufficient outcome-neutral conditions (e.g. positive controls) for ensuring that the results obtained are able to test the stated hypotheses

Not really applicable here?

Reviewer: 3

Comments to the Author(s)

This pre-registered study proposes to identify the psychological factors that are most strongly associated with increased engagement in real-world collective climate action as a result of a 12-week intervention. Rather than testing a specific theory about the link between psychological factors and action, the authors propose two novel methodological tools: a video that attempts to boost 11 identified psychological factors and a behavior-tracking methodology to measure real-world collective action. The proposed single-cohort pre-post longitudinal study will recruit 160 participants aged 18-30 who will complete an online survey assessing the 11 psychological factors of interest, as well as self-reporting baseline activism behaviors and demographics. They will complete a 12-week video-based intervention, during which their activism will be monitored. At the end of the 12 weeks, they will complete the questionnaires for a second time. Proposed analyses will assess the pre-post change in the psychological factors, in behavioral outcomes (objectively monitored and self-reported), and the relationship between these changes.

The research question is timely and important and is well-motivated by the introduction. The methodology does a good job of outlining the design, the sample size, procedures related to outliers, and proposed analyses, including exploratory analyses - though some key gaps, including specification of DVs, need to be addressed, as outlined below. The issue of power is addressed. The proposed methodology and analysis pipeline appear largely sound and feasible, and are able to test the stated hypotheses, though I have some specific questions and concerns, which are outlined below. If these concerns are addressed, the project is likely to generate a rich

and important dataset that will demonstrate whether video-based interventions can boost collective climate action, and will give new insights into the psychological factors that contribute to such change.

1. In the abstract and introduction, the authors highlight a lack of causal evidence regarding the psychological factors that predict collective climate action. A key drawback of the proposed single-cohort pre-post design, however, is that the absence of a control group will preclude causal inferences. While I sympathize with the arguments made regarding the sample size and cost entailed by the inclusion of a control group, the fact that the lack of a control group weakens claims to causal links should be acknowledged and, as a result the authors should perhaps rethink their focus on causal evidence.
2. The 11 psychological factors that will be investigated are not well motivated in the introduction - there is general reference to psychological factors rather than specific reference to the 11 that will be examined here. The rationale for these specific 11 factors should be expanded.
3. Further, it is unclear how the questionnaire that assesses these 11 factors was developed - it appears that questions were selected from previous studies and instruments, but how this was done, and whether any effort has been made to pilot this instrument is not reported. There is very unequal treatment of the 11 factors in the questionnaire - each factor is assessed with a different number of items (e.g., some factors are assessed with three questions, some with as many as 16) and using different types of response scales (some -3 to 3; some 1 to 7). The variable psychometric characteristics could pose a significant confound for any attempt to identify the factors that contribute to action. It is not clear why there are such differences, what their potential impact is, and how they will be controlled for or any potential impact mitigated against - other than the proposal to exclude questions that reduce Cronbach's alpha or to select "a single face-valid item".
4. Further details on the video intervention are required. First - as is the case for the questionnaire, it is unclear how the video intervention was developed and whether any attempt has been made to validate it - for example to assess the extent to which the 11 psychological factors are present or evoked by the video. It is not clear whether the participants will see the same or a different video each week. If the video is different, how will the quality/extent to which these factors are targeted be controlled across weeks? Will order of presentation be counterbalanced across participants?
5. Another shortcoming of existing research highlighted by the authors is the reliance on self-report or intentions rather than objectively measured participation - the authors proposed a novel "behavior-tracking methodology to measure real-world participation in climate action." This methodology is scantily described, however. It is not clear how the "partner organizations will monitor how many climate activism events are attended by each participant." This seems practically and ethically challenging to do and further details are needed. It seems likely that the authors will ultimately have to rely on self-reported participation, which again weakens the claims regarding novelty and advance on previous work. If the behavior-tracking methodology is successful, what will be measured, over what time scale, and how this will be included as an outcome variable in the analyses is not specified.
6. In the section on page 11 addressing outcome variables, climate activism events attended are not included as an outcome variable. As mentioned in the previous comment, it is not clear how the objective measure will be computed.
7. How will demand characteristics be addressed?
8. The study has the potential to generate a very rich and valuable dataset - a further potential exploratory analysis might include factor analyses of the questionnaire data, which might aid in data reduction, and in the identification of the most important factors/families of factors.

Author's Response to Decision Letter for (RSOS-210006.R0)

See Appendix B.

RSOS-210006.R1 (Revision)

Review form: Reviewer 2

Do you have any ethical concerns with this paper?

No

Recommendation?

Major revision

Comments to the Author(s)

After looking at the comments to the reviewers and the revised manuscript, I think the mentioned concerns have been addressed well.

I like how a baseline-phase was included, which helps with many original concerns and improves the overall quality of the study and the foundation of the causal interpretation. I was particularly impressed how the potential concern that participants attend in events other than the partner climate organizations' was anticipated and addressed. Several issues with transparency and the wording of the hypotheses have been addressed, as well.

The only thing that came to my mind reading the manuscript now was the exclusion of participants who attended more than one climate activism event. As mentioned before, I like the rationale for choosing this participant restriction. Nonetheless, I think it might be helpful to acknowledge/ mention more explicitly that only a somewhat specific sample of participants (resulting from the integration of the screening questions) will be collected (i.e. participants who believe in anthropogenic climate change but have not previously engaged in climate activism). I think this is an important subgroup to target (and maybe even the subgroup with the biggest potential for being mobilized), but it might be helpful to acknowledge that this is still a subgroup of the population regarding factors such as climate change beliefs and climate activism. However, this is no concern about methodological challenges but just an issue of wording.

One other small point I noticed is in Page 7, Line 27, where the text says, "Participants will undergo an initial screening for age (18-30), native English, (...)". Judging from the comments to reviewer #1, I assume that this might be a leftover from the previous version of the manuscript and the requirement for participants to have spoken English from childhood on ought to be removed?

Overall, the study sounds very interesting and I am excited about the commitment to study "actual" climate activism behavior instead of relying only on questionnaires. I am looking forward to learning about the results of the study!

Review form: Reviewer 3

Do you have any ethical concerns with this paper?

No

Recommendation?

Accept in principle

Comments to the Author(s)

The authors have comprehensively addressed my concerns - I wish them the very best of luck with the study.

Decision letter (RSOS-210006.R1)

Dear Dr Castiglione,

On behalf of the Editors, I am pleased to inform you that your Manuscript RSOS-210006.R1 entitled "Discovering the psychological building blocks underlying climate action - a longitudinal study of real-world activism" has been accepted in principle for publication in Royal Society Open Science subject to minor revision in accordance with the referee and editor suggestions. Please find their comments at the end of this email.

The reviewers and handling editors have recommended publication, but also suggest some minor revisions to your manuscript. Therefore, I invite you to respond to the comments and revise your manuscript.

Please you submit the revised version of your manuscript within 7 days (i.e. by the 18-Mar-2021). If you do not think you will be able to meet this date please let me know immediately.

When submitting your revised manuscript, you will be able to respond to the comments made by the referees and you should upload a file "Response to Referees". You can use this to document any changes you make to the original manuscript. In order to expedite the processing of the revised manuscript, please be as specific as possible in your response to the referees.

Full author guidelines can be found here <https://royalsocietypublishing.org/rsos/registered-reports>.

on behalf of Professor Chris Chambers (Subject Editor, Royal Society Open Science)
openscience@royalsociety.org

Associate Editor Comments to Author (Professor Chris Chambers):

Two of the three original reviewers were available to review the manuscript on a short timeframe, and at the outset I would like to express my sincere gratitude to both reviewers for the speedy assessments.

The good news is that both reviewers are now satisfied with the manuscript, although Reviewer 2 notes a couple of final areas that would benefit from fine tuning before in-principle acceptance (IPA) is awarded. Please respond to these points in a final revision, and IPA will then be forthcoming without requiring further in-depth Stage 1 review.

Reviewer comments to Author:

Reviewer: 2

Comments to the Author(s)

After looking at the comments to the reviewers and the revised manuscript, I think the mentioned concerns have been addressed well.

I like how a baseline-phase was included, which helps with many original concerns and improves the overall quality of the study and the foundation of the causal interpretation. I was particularly impressed how the potential concern that participants attend in events other than the partner climate organizations' was anticipated and addressed. Several issues with transparency and the wording of the hypotheses have been addressed, as well.

The only thing that came to my mind reading the manuscript now was the exclusion of participants who attended more than one climate activism event. As mentioned before, I like the rationale for choosing this participant restriction. Nonetheless, I think it might be helpful to acknowledge/ mention more explicitly that only a somewhat specific sample of participants (resulting from the integration of the screening questions) will be collected (i.e. participants who believe in anthropogenic climate change but have not previously engaged in climate activism). I think this is an important subgroup to target (and maybe even the subgroup with the biggest potential for being mobilized), but it might be helpful to acknowledge that this is still a subgroup of the population regarding factors such as climate change beliefs and climate activism. However, this is no concern about methodological challenges but just an issue of wording.

One other small point I noticed is in Page 7, Line 27, where the text says, "Participants will undergo an initial screening for age (18-30), native English, (...)". Judging from the comments to reviewer #1, I assume that this might be a leftover from the previous version of the manuscript and the requirement for participants to have spoken English from childhood on ought to be removed?

Overall, the study sounds very interesting and I am excited about the commitment to study "actual" climate activism behavior instead of relying only on questionnaires. I am looking forward to learning about the results of the study!

Reviewer: 3

Comments to the Author(s)

The authors have comprehensively addressed my concerns - I wish them the very best of luck with the study.

Author's Response to Decision Letter for (RSOS-210006.R1)

See Appendix C.

Decision letter (RSOS-210006.R2)

Dear Dr Castiglione

On behalf of the Editor, I am pleased to inform you that your Manuscript RSOS-210006.R2 entitled "Discovering the psychological building blocks underlying climate action - a longitudinal study of real-world activism" has been accepted in principle for publication in Royal Society Open Science.

You may now progress to Stage 2 and complete the study as approved. Before commencing data collection we ask that you:

- 1) Update the journal office as to the anticipated completion date of your study.
- 2) Register your approved protocol on the Open Science Framework (<https://osf.io/>) or other recognised repository, either publicly or privately under embargo until submission of the Stage 2 manuscript. Please note that a time-stamped, independent registration of the protocol is mandatory under journal policy, and manuscripts that do not conform to this requirement cannot be considered at Stage 2. The protocol should be registered unchanged from its current approved state, with the time-stamp preceding implementation of the approved study design. **We strongly recommend using the dedicated registration portal for Stage 1 RRs at <https://osf.io/rr> which only takes a few minutes and results in the protocol being entered into the OSF Registry**

Following completion of your study, we invite you to resubmit your paper for peer review as a Stage 2 Registered Report. Please note that your manuscript can still be rejected for publication at Stage 2 if the Editors consider any of the following conditions to be met:

- The results were unable to test the authors' proposed hypotheses by failing to meet the approved outcome-neutral criteria.
- The authors altered the Introduction, rationale, or hypotheses, as approved in the Stage 1 submission.
- The authors failed to adhere closely to the registered experimental procedures. Please note that any deviations from the approved experimental procedures must be communicated to the editor immediately for approval, and prior to the completion of data collection. Failure to do so can

result in revocation of in-principle acceptance and rejection at Stage 2 (see complete guidelines for further information).

- Any post-hoc (unregistered) analyses were either unjustified, insufficiently caveated, or overly dominant in shaping the authors' conclusions.
- The authors' conclusions were not justified given the data obtained.

We encourage you to read the complete guidelines for authors concerning Stage 2 submissions at <https://royalsocietypublishing.org/rsos/registered-reports#ReviewerGuideRegRep>. Please especially note the requirements for data sharing, reporting the URL of the independently registered protocol, and that withdrawing your manuscript will result in publication of a Withdrawn Registration.

Once again, thank you for submitting your manuscript to Royal Society Open Science and we look forward to receiving your Stage 2 submission. If you have any questions at all, please do not hesitate to get in touch. We look forward to hearing from you shortly with the anticipated submission date for your stage two manuscript.

on behalf of Professor Chris Chambers (Registered Reports Editor, Royal Society Open Science)
openscience@royalsociety.org

Author's Response to Decision Letter for (RSOS-210006.R2)

See Appendix D.

RSOS-210006.R3

Review form: Reviewer 1

Is the manuscript scientifically sound in its present form?

Yes

Are the interpretations and conclusions justified by the results?

Yes

Is the language acceptable?

Yes

Do you have any ethical concerns with this paper?

No

Have you any concerns about statistical analyses in this paper?

Yes

Recommendation?

Accept with minor revision

Comments to the Author(s)

The manuscript is sound, my only comment is that the authors claim to have completed a thematic analysis in study, however, the qualitative analysis appears to be a summative content analysis not a thematic analysis. I recommend the authors update how they have labelled the analysis. Here is a suggested paper for describing content analysis.
<https://journals.sagepub.com/doi/abs/10.1177/1049732305276687>

Review form: Reviewer 2

Is the manuscript scientifically sound in its present form?

Yes

Are the interpretations and conclusions justified by the results?

Yes

Is the language acceptable?

Yes

Do you have any ethical concerns with this paper?

No

Have you any concerns about statistical analyses in this paper?

Yes

Recommendation?

Accept with minor revision

Comments to the Author(s)

This manuscript deals with a very important and pressing issue – climate activism. I furthermore think the study makes a meaningful contribution to the literature by using and suggesting an interesting way to measure actual activism behaviour, as well as reporting interesting results. However, I recommend some revisions involving minor edits, as well as revisions regarding how the statistical analyses are reported, and the accessibility of the data. Please see the attached review letter for more details (Appendix E).

Review form: Reviewer 3

Is the manuscript scientifically sound in its present form?

Yes

Are the interpretations and conclusions justified by the results?

Yes

Is the language acceptable?

Yes

Do you have any ethical concerns with this paper?

No

Have you any concerns about statistical analyses in this paper?

No

Recommendation?

Accept with minor revision

Comments to the Author(s)

The paper by Castiglione et al. is a Stage 2 Registered report. The authors have done an excellent job of implementing and reporting on the study as proposed, as well as conducting an important exploratory replication. While the results were disappointing, the study provides valuable insights that will benefit future research. In particular, the analysis of semi-structured interview data to understand what held back the participants from engaging in the activism events, and the lessons emerging from the in-person replication, which also failed to increase engagement in activism, highlight the challenge of fostering collective action and will stimulate further work.

Regarding the results - a fuller presentation of the data and results of analyses carried out for H1 and H2 is required. Ideally, a table should provide means and standard deviations, t-statistics, p-values and effect sizes for each of the 11 psychological factors and three behavioural outcomes examined. Alternatively, the data could be presented in graphical form, as in Figure 2 (though please see comment below).

A second comment is that I found the general discussion very short. The brief discussion seems like a missed opportunity to engage with the question of how the successfully induced increases in affective engagement and self- and collective-efficacy (with the latter two theorised to constitute key factors in individual and collective action) might be converted into active participation. A second important observation was the fact that the pressures of modern student life (juggling college-work plus work-work) form a significant barrier to engagement in activism, despite an awareness of and concern about climate. How to overcome the near-universal barrier of time poverty may be one of the most important questions we should examine.

Minor comment - in Figure 2, a better way to present the data would be to use separate graphs for each psychological factor and to present "post" and "pre" side-by-side (i.e., use hue for timepoint, not factor).

Decision letter (RSOS-210006.R3)

Dear Dr Castiglione:

On behalf of the Editor, I am pleased to inform you that your Stage 2 Registered Report RSOS-210006.R3 entitled "Discovering the psychological building blocks underlying climate action - a longitudinal study of real-world activism" has been deemed suitable for publication in Royal Society Open Science subject to minor revision in accordance with the referee suggestions. Please find the referees' comments at the end of this email.

The reviewers and Subject Editor have recommended publication, but also suggest some minor revisions to your manuscript. We invite you to respond to the comments and revise your manuscript. Below the referees' and Editors' comments (where applicable) we provide additional requirements. Final acceptance of your manuscript is dependent on these requirements being met. We provide guidance below to help you prepare your revision.

Please submit your revised manuscript and required files (see below) no later than 21 days from today's (ie 2nd May) date. Note: the ScholarOne system will 'lock' if submission of the revision is attempted 21 or more days after the deadline. If you do not think you will be able to meet this deadline please contact the editorial office immediately.

on behalf of Professor Chris Chambers (Associate Editor) and Chris Chambers
(Registered Reports Editor, Royal Society Open Science)
openscience@royalsociety.org

Associate Editor Comments to Author (Professor Chris Chambers):

Associate Editor: 1

Comments to the Author:

The three reviewers who assessed your Stage 1 proposal kindly returned to evaluate the completed Stage 2 manuscript. As you will, see all of the assessments are broadly very positive and the manuscript is nearly suitable for publication. There are, however, a range of minor-to-moderate issues to address concerning the depth of consideration in the discussion, reporting of results, and data availability. In revising, please do not make any changes to the accepted Stage 1 part of the manuscript unless doing so is necessary to correct an error or avoid a misunderstanding. I will assess a revised manuscript at desk, and provided you are able to address these points comprehensively in a revision and response, full acceptance should be forthcoming without requiring further in-depth review.

Comments to Author:

Reviewer: 3

Comments to the Author(s)

The paper by Castiglione et al. is a Stage 2 Registered report. The authors have done an excellent job of implementing and reporting on the study as proposed, as well as conducting an important exploratory replication. While the results were disappointing, the study provides valuable insights that will benefit future research. In particular, the analysis of semi-structured interview

data to understand what held back the participants from engaging in the activism events, and the lessons emerging from the in-person replication, which also failed to increase engagement in activism, highlight the challenge of fostering collective action and will stimulate further work.

Regarding the results - a fuller presentation of the data and results of analyses carried out for H1 and H2 is required. Ideally, a table should provide means and standard deviations, t-statistics, p-values and effect sizes for each of the 11 psychological factors and three behavioural outcomes examined. Alternatively, the data could be presented in graphical form, as in Figure 2 (though please see comment below).

A second comment is that I found the general discussion very short. The brief discussion seems like a missed opportunity to engage with the question of how the successfully induced increases in affective engagement and self- and collective-efficacy (with the latter two theorised to constitute key factors in individual and collective action) might be converted into active participation. A second important observation was the fact that the pressures of modern student life (juggling college-work plus work-work) form a significant barrier to engagement in activism, despite an awareness of and concern about climate. How to overcome the near-universal barrier of time poverty may be one of the most important questions we should examine.

Minor comment - in Figure 2, a better way to present the data would be to use separate graphs for each psychological factor and to present "post" and "pre" side-by-side (i.e., use hue for timepoint, not factor).

Reviewer: 2

Comments to the Author(s)

This manuscript deals with a very important and pressing issue – climate activism. I furthermore think the study makes a meaningful contribution to the literature by using and suggesting an interesting way to measure actual activism behaviour, as well as reporting interesting results. However, I recommend some revisions involving minor edits, as well as revisions regarding how the statistical analyses are reported, and the accessibility of the data. Please see the attached review letter for more details.

Reviewer: 1

Comments to the Author(s)

The manuscript is sound, my only comment is that the authors claim to have completed a thematic analysis in study, however, the qualitative analysis appears to be a summative content analysis not a thematic analysis. I recommend the authors update how they have labelled the analysis. Here is a suggested paper for describing content analysis.

<https://journals.sagepub.com/doi/abs/10.1177/1049732305276687>

===PREPARING YOUR MANUSCRIPT===

one version should clearly identify all the changes that have been made (for instance, in coloured highlight, in bold text, or tracked changes);

===PREPARING YOUR REVISION IN SCHOLARONE===

- If you are providing image files for potential cover images, please upload these at this step, and inform the editorial office you have done so. You must hold the copyright to any image provided.
- A copy of your point-by-point response to referees and Editors. This will expedite the preparation of your proof.

- Ensure that your data access statement meets the requirements at <https://royalsociety.org/journals/authors/author-guidelines/#data>. You should ensure that you cite the dataset in your reference list. If you have deposited data etc in the Dryad repository, please only include the 'For publication' link at this stage. You should remove the 'For review' link.
- If you are requesting an article processing charge waiver, you must select the relevant waiver option (if requesting a discretionary waiver, the form should have been uploaded, see 'File upload' above).
- If you have uploaded any electronic supplementary (ESM) files, please ensure you follow the guidance at <https://royalsociety.org/journals/authors/author-guidelines/#supplementary-material> to include a suitable title and informative caption. An example of appropriate titling and captioning may be found at https://figshare.com/articles/Table_S2_from_Is_there_a_trade-off_between_peak_performance_and_performance_breadth_across_temperatures_for_aerobic_scope_in_teleost_fishes_/3843624.

Author's Response to Decision Letter for (RSOS-210006.R3)

See Appendix F.

Decision letter (RSOS-210006.R4)

Dear Dr Castiglione:

It is a pleasure to accept your manuscript entitled "Discovering the psychological building blocks underlying climate action - a longitudinal study of real-world activism" in its current form for publication in Royal Society Open Science.

Thank you for your fine contribution. On behalf of the Editors of Royal Society Open Science, we look forward to your continued contributions to the journal.

on behalf of Professor Chris Chambers (Subject Editor)
openscience@royalsociety.org

Appendix A

Discovering the psychological building blocks underlying climate action - a longitudinal study of real-world activism

Congratulations on studying the mechanisms of activism. Activism is an often-overlooked category of pro-environmental behavior, with huge potential to lead to large scale system change.

General comments:

Please provide a clearer definition of climate activism including what it does not include. In your description of activism behaviors in your introduction, you described activism as: Financial support, circulation of petitions, protest participation and joining events organized by ecological groups. In your measurement of activism, you do not include financial support but rather include this in environmental education and leadership behaviors. There needs to be a clearer delineation of what makes an activism behavior vs an educational or leadership behavior, and that these behaviors together make up activism.

Please acknowledge that the types of people who will be able to engage in your intervention will be educated, predominantly white, English speaking and any other factors which make them different to the broader population.

It is unclear why participants need to have spoken English from childhood. If participants are students studying at a university level, they should have the English competency to participate in this experiment.

Specific comments:

Page 4, Line 54

“The above findings have emerged in diverse populations”

I am not convinced that these populations are particularly diverse. The nations listed are all predominantly white, developed western nations, highly individualistic, masculine cultures with low power distance. Please highlight what makes them diverse, perhaps refer to differences in the key psychological variables you will be using.

Page 7, Line 15

Fantastic that you will be exploring your findings and adding richness through the use of interviews! In my experience, Kappa is not particularly useful unless using for very clear themes or content codes. It also seems unnecessary and expensive for both reviewers to code every interview. Could you get two coders to code the first two interviews and discuss discrepancies, then revise the coding. Then either coder could continue with the remainder of the interviews.

Can you add some alternative information describing how you will ensure the quality of the interviews? I would like to see an interview summary returned to the participants for member checking.

How will the participants for the semi-structured interviews be selected? If you end up with the majority of participants from the survey consenting to participation in the interview will you interview all participants? If not, how will you ensure the interviews are representative of the participant group?

Will you use any specific software to code the interviews?

Will you use any specific Thematic Analysis steps, (e.g., Braun and Clarke, 2006).

Page 24, Line 40

I recommend adding a “not applicable” response option to the Emission Reduction Behaviors scale. For example, If someone does not drive then they can not provide a valid response to the transport questions. You might also ask these questions a second time and add “How often do you engage in the following behavior for **environmental reasons?**”

Page 30, Line 10

Very happy to see the inclusion of attention traps. What will the cut off score be for a participant to be excluded from the study?

Appendix B

Dear Editor,

We were very gratified that Reviewer #1 was pleased with our research topic on the mechanisms of climate activism, that Reviewer #2 said of our preregistered study plan that "... the research question is well-founded in existing scientific research. Additionally, I think the research question has important practical implications," and that Reviewer #3 wrote "The research question is timely and important and is well-motivated by the introduction. The methodology does a good job of outlining the design, the sample size, procedures related to outliers, and proposed analyses, including exploratory...The proposed methodology and analysis pipeline appear largely sound and feasible, and are able to test the stated hypotheses"

Below we address each of their points one by one. We feel we can respond to all the points with minimum changes: the only major change is we have followed Reviewer #2's recommendation to add a baseline period to the study (6 weeks, see below).

We thank you for your efforts and we thank the reviewers for their helpful and perceptive comments. Frankly, these are some of the best-spirited and most constructive reviewers' comments we've ever seen.

We feel the paper is now stronger and we hope you find it acceptable for the journal.

We have a special request: could this round be expedited? We have added a baseline control period, as per one of the Reviewer's suggestions, and, even when squeezing the study into 3 months (1.5 baseline, 1.5 intervention) it is a big challenge for us to finish it by finals week in June. Once we hit finals week the students will be too distracted to participate. And we would not be able to run the study until end of September when they 'return'. That pushes us beyond the lifetime of the funded project. If the reviewers can promptly review, we still have a chance to run this study before summer.

Sincerely,
Anna Castiglione, Adam Aron, Cameron Brick, Stefanie Holden and Debra Lindsay

Comments to Author:

Reviewer: 1

Comments to the Author(s)

Congratulations on studying the mechanisms of activism. Activism is an often-overlooked category of pro-environmental behavior, with huge potential to lead to large scale system change. I recommend minor revisions such that there are clearer definitions of key terms, and more appropriate quality assurance techniques for the qualitative component.

General comments:

Please provide a clearer definition of climate activism including what it does not include. In your description of activism behaviors in your introduction, you described activism as: Financial support, circulation of petitions, protest participation and joining events organized by ecological groups. In your measurement of activism, you do not include financial support but rather include this in environmental education and leadership behaviors. There needs to be a clearer delineation of what makes an activism behavior vs an educational or leadership behavior, and that these behaviors together make up activism.

We have now clarified our definition of environmental activism (page 2, line 36):

"Together, this research has identified a dozen psychological factors that relate to self-reported environmental activism, where this includes joining events organized by ecologically-oriented groups, and engaging in environmental education and leadership behaviors (such as circulation of petitions, raising awareness about climate issues, outreach and community organizing)."

Please acknowledge that the types of people who will be able to engage in your intervention will be educated, predominantly white, English speaking and any other factors which make them different to the broader population.

Good point. We can use typical base rates for our studies at UCSD. We have now added (page 7, line 18):

“We acknowledge that given the university undergraduate population from which we will draw our sample, the participants will be predominantly White and Latinx, and first language English speaking.”

It is unclear why participants need to have spoken English from childhood. If participants are students studying at a university level, they should have the English competency to participate in this experiment.

This requirement was introduced to ensure participants would be able to comprehend the videos well enough, but we agree with the reviewer that university students attending English courses should be able to have a high level of comprehension of the English language already. We have now removed this requirement.

Specific comments:

Page 4, Line 54

“The above findings have emerged in diverse populations”

I am not convinced that these populations are particularly diverse. The nations listed are all predominantly white, developed western nations, highly individualistic, masculine cultures with low power distance. Please highlight what makes them diverse, perhaps refer to differences in the key psychological variables you will be using.

We have added the following acknowledgement (page 3 line 4):

“The above findings have emerged in diverse geographic locations (although predominantly white, developed western nations were studied), from Germany¹¹, Austria¹⁸, the USA^{14,13}, Australia¹⁷, and other countries¹³.”

Page 7, Line 15

Fantastic that you will be exploring your findings and adding richness through the use of interviews! In my experience, Kappa is not particularly useful unless using for very clear themes or content codes. It also seems unnecessary and expensive for both reviewers to code every interview. Could you get two coders to code the first two interviews and discuss discrepancies, then revise the coding. Then either coder could continue with the remainder of the interviews.

We thank the reviewer for this suggestion. We have modified our rating plan as follows (page 16, line 27):

“Following transcription, participants will be contacted with a copy of the interview transcript for review and approval. We will then conduct a thematic analysis of all transcripts, to identify common themes using NVivo software. We will adhere to the guidelines of thematic analysis³⁵. Two coders will start by familiarizing themselves with the data and collaborating over the generation of initial thematic codes. Following the establishment of the codebook, they will each code two specific transcripts (i.e. the same ones), discuss discrepancies, and revise the codebook as necessary. The remaining transcripts will be split between the two coders. Upon completion of the coding process, they will then work collaboratively to identify, review, and define the themes that arise.”

Can you add some alternative information describing how you will ensure the quality of the interviews? I would like to see an interview summary returned to the participants for member checking.

We thank the reviewer for their suggestion and have modified the plan by adding the following (page 16, line 20):

“Interviews will be conducted through online video via Zoom and will last one hour. Audio of the interviews will be recorded and transcribed for analysis, removing any identifiable information, and then destroyed. Interviews will be coded and transcribed by NVivo Software. Following transcription, participants will be contacted with a copy of the interview transcript for review and approval. We will then conduct a thematic analysis of all transcripts, to identify common themes using NVivo software. We will adhere to the guidelines of thematic analysis.”

How will the participants for the semi-structured interviews be selected? If you end up with the majority of participants from the survey consenting to participation in the interview will you interview all participants? If not, how will you ensure the interviews are representative of the participant group?

We have now specified (page 16, line 15):

“**Semi-structured interview.** We will interview those participants who express their interest in this part of the study. If the number is large, we will randomly select a subset to keep the interview schedule tractable.”

Will you use any specific software to code the interviews?

We have now specified (page 16, line 24):

“Interviews will be coded and transcribed using NVIVO Software.”

Will you use any specific Thematic Analysis steps, (e.g., Braun and Clarke, 2006).

We have now specified (page 16, line 31):

“We will adhere to the guidelines of thematic analysis³⁵. Two coders will start by familiarizing themselves with the data and collaborating over the generation of initial thematic codes. Following the establishment of the codebook, they will each code two specific transcripts (i.e. the same ones), discuss discrepancies, and revise the codebook as necessary. The remaining transcripts will be split between the two coders. Upon completion of the coding process, they will then work collaboratively to identify, review, and define the themes that arise.”

Page 24, Line 40

I recommend adding a “not applicable” response option to the Emission Reduction Behaviors scale. For example, If someone does not drive then they can not provide a valid response to the transport questions. You might also ask these questions a second time and add “How often do you engage in the following behavior for environmental reasons?”

We thank the reviewer for these helpful suggestions. We have now added a “Not applicable” answer to the Emission Reduction Behaviors survey, and we have added “for environmental reasons” to the question (page 24, line 40).

Page 30, Line 10

Very happy to see the inclusion of attention traps. What will the cut off score be for a participant to be excluded from the study?

We have now specified an exclusion criteria (page 13, line 13):

“Exclusion criteria

Participants missing or answering incorrectly to more than 10 questions (2 bulletins per week, 2 questions per bulletin, for a total of 24 questions) during the baseline period (these relate to the event bulletin only) or more than 24 attention check questions (2 videos of 5 questions each + 2 bulletins of 2 questions each per week, for 6 weeks, for a total of 84 questions) during the

intervention period (these relate to the videos and to the event bulletin) will be excluded from the analysis.”

Reviewer: 2

Comments to the Author(s)

1. The scientific validity of the research question(s)

I think the research question is well-founded in existing scientific research. The psychological factors that are chosen are frequently reported to play a role in predicting climate activism, while interesting relatively new ones (e.g. Theory of change) are included, too. Additionally, I think the research question has important practical implications. From what I understood, the idea was to find out which of the 11 psychological factors are more and which are less important in causing environmental activism. I think answering this question can be very important in applied climate activism, in order to know how to best use limited resource for mobilization. Furthermore, I think a video intervention is especially practical regarding the low costs, but also regarding the high feasibility during the covid-19 pandemic. A particular strength of the article is the aim to measure climate activism through a behavioral, “objective” measure, instead of relying on self-reported climate activism. Moreover, including environmental education and leadership behaviors as outcome variables could provide interesting new nuances. Including semi-structured interviews could furthermore be a very meaningful measure to find out the strengths and limitation of the intervention.

2. The logic, rationale, and plausibility of the proposed hypotheses

H1: “Some of the 11 psychological factors related to climate activism will increase from baseline to follow-up.”

While this is a valid research question, **this does not constitute a testable or falsifiable hypothesis.** This hypothesis would be confirmed as soon as one factor increases (non-significantly) from baseline to follow-up, even if all other factors don’t increase or even decrease. The hypothesis should be made more specific in order to prevent undisclosed flexibility in the interpretation of results. With the current wording, what constitutes a confirmation of the hypothesis is greatly open to interpretation by the researchers. The same applies to H3.

The reviewer is correct in that our study is non-specific. We don’t have a theory about *which* of the 11 psychological factors (drawn from the extant literature), counts, but our aim is to discover which do. H1 is better stated as “one or more” of the factors will increase. And we specified that we will use multiple comparisons corrections. We have now reworded H1 and H3 to the following (page 13, line, 29):

“**H1:** We expect that one or more of the 11 psychological factors related to climate activism will increase from baseline to follow-up. *

H2: Each behavioral outcome will increase from baseline to follow-up: a) climate activism events attended (objective, self-reported); b) emissions-reduction behaviors, and c) environmental education and leadership behaviors.

H3: For each behavioral outcome, any change from baseline will be explained by changes in one or more of the psychological factors from beginning to end of the study.”

Logic:

- Is the factor Faith in Institutions expected to increase, too? The video intervention could lead to an increase and to a decrease of faith in institutions (intervention: “a combination of the history of unproductive international and national climate policy initiatives, combined with examples of current politicians that have an earnest climate crisis focus”).

We thank the reviewer for pointing out this important issue. “Faith in institutions” is a particularly complex factor. While some literature has suggested that *increased* trust in policy elites increases people’s motivation for environmental activism, other considerations suggest that *decreased* trust in such policy

elites may increase their activism (i.e. when people comprehend the scale of governmental inaction on climate of the past few decades, especially emphasized by our videos, they may be moved to join grassroots action). We therefore prefer to maintain neutral expectations towards how our intervention will modulate this factor and how changes in this factor will affect participation, and we now have specified so in the paper (page 13, footnote).

“Factor 3 (“Faith in institutions”) is a complex factor that could be pushed up or down. For example, some participants may conclude from the material we show that we simply cannot rely on policy elites (given the general failure of UN and governmental policy over 30 years); but others might feel more faith in institutions when they see, for example, how the Sunrise Movement boosted congressional action. We therefore maintain neutral expectations how this factor may vary and how this variation may affect participation”

- What does it mean if the factor “theory of change” increases? Politicians or people? Could be more specific.

We have now added a more thorough explanation of what “theory of change” is, and how this factor may be involved in activism (Page 9, line 43)

“This last factor is a novel measure of participant’s belief on how successful change would be implemented: either as a bottom-up process stemming from grassroots action and expanding to the whole society, or as a top-down process implemented by governments and authorities. We expect that belief in bottom-up change will more strongly increase the motivation to engage in action.”

H2: “Each behavioral outcome will increase from baseline to follow-up: a) climate activism events attended (objective, self-reported)”

For objective climate activism events attended, from my understanding, there is no baseline measurement, thus, it can’t increase from baseline to follow-up. Otherwise, the objective climate activism measure is not included in any hypothesis, while (judging from the introduction) it is supposed to be a central point of the current study. What are the plans for interpreting and analyzing this measure? I think, trying to measure climate activism more objectively is a great idea. However, I was wondering what the plan for interpretation is, if self-report and objective measures differ? What would speak more towards a confirmation of the hypothesis? What will be regarded as the more valid measure to draw conclusions from? In order to avoid being able to change what constitutes a confirmation of the hypothesis after having analyzed the results, **it should be specified in advance, which measure is regarded as the most important outcome measure.**

We thank the reviewer for this valuable feedback; we have revisited our baseline measure of event attendance by introducing – as suggested by the reviewer – a pre-intervention baseline period (see point 3. below). We now have also specified that objective event attendance is our main measure of activism (page 5, line 15):

“The main measure of activism behaviour in this study is event participation (objectively ratified). We will embed a study-team member in the UCSD GND and SD350 climate groups and have her verify participation in events held by both organizations. Additionally, self-reported environmental education and leadership behaviour will be collected, along with other pro-environmental behaviours relevant to climate change..”

3. The soundness and feasibility of the methodology and analysis pipeline (including statistical power analysis where applicable)

No pre-/ baseline measure for “objective” activism behavior:

In the introduction, it is clearly noted that “noise and bias in self-report measures of activism behavior” exists and it is important to assess activism behavior objectively, instead of relying on self-report. I really like that idea and think it is a very interesting approach to look at actual participation in activism events. However, it strikes me as unusual then, that the study protocol does not include (or at least does not report) a baseline measure of “objective” activism behavior. If the study protocol does include a baseline

measure of objective activism behavior, I strongly encourage to make this more explicit in the methods. If no baseline measurement of objective activism behavior is included, this strongly limits the interpretation with regards to the effectiveness of the intervention. Including a baseline measurement **could be done by including a (equally long) baseline-phase, where participants receive no video intervention or booklet, (or only receive the booklet about activism events that happen this week,) but “objective” attendance to activism events is still recorded.** However, not including such a baseline measure does not allow for causal inferences about objective activism behavior. It is a great goal to measure activism behavior “objectively”, but if there is no data on objective activism behavior available from before the intervention, the analyses for the effectiveness of the intervention will have to rely again on self-reports of activism behavior.

We have now added a 6-week baseline period before the video intervention (which itself is now shortened to 6 weeks, in order to be able to run the whole experiment in 3 months while adding a baseline period). We have adapted our plan to the following: 1) participants will be screened for participation 2) **Baseline**: they will undergo 6-weeks where they receive a climate bulletin only, and no video intervention 3) they will take a baseline survey (assessing the 11 psychological factors and the two self-reported behavioral outcomes), 3) **Intervention**: they will undergo the video intervention plus receive the event bulletins, 4) they will take the final survey and 5) they will undergo the semi-structured interview. We explain these steps in the paper, and specify the baseline period as follows (page 8, line 10):

“Baseline Period (6 weeks)

For the first 6 weeks, the Qualtrics surveys received by the participants will contain a "climate events bulletin," consisting of a list of events happening in the next 3-4 days, held by our partner climate organizations (San Diego 350 and UCSD Green New Deal) (**Fig.1B**). At the top of the bulletin, participants will be reminded that, while they are required to scroll to the end of the list, they are not required to attend these events (there will be no monetary penalty for not attending). To ensure that participant read the event bulletin, they will be asked two attention-check questions at the end of each bulletin, asking about details of the events. Responding incorrectly to more than one question will potentially result in a \$3 penalty for that overall session (\$3 will be subtracted from the final payment of \$100); however, the subject will be directed to read the bulletin again and to re-take the final test to avoid losing the \$3 (i.e. they are allowed two attempts).“

Additionally, in order to reduce the confound of event participation at organizations other than our partners' (which would be hard to track and incorporate in our assessment of event participation across the study), we will exclude any participant who has ever participated in more than one activist event (e.g. a protest, or campaign of any kind). We specify this additional screening criterion in the paper (page. 7, line 36):

“A concern in our design, is that some participants might attend events of climate organizations other than UCSD GND and SD350 (and we won't be able to tell). To obviate this concern, we will ask everyone at recruitment if they have ever attended an event for an environmental organization (such as a protest). Those answering yes will then be asked how many times they attended such an event (once, more than once). Participants responding “more than once” will be excluded from the study.”

Confounding variable systematically varying with the intervention: climate events bulletin?

I was wondering whether the climate events bulletin was supposed to be part of the intervention, or whether the video intervention was assumed to work independently of the climate events bulletin? **If so, a baseline phase for the bulletin only (without the video intervention) would be desirable, in order to be able to distinguish the effect of the event bulletin from the effect of the video intervention.** It could be possible that all observed changes in psychological factors and behavioral outcomes are solely the result of reading the climate events bulletin. For example, making it salient through the bulletin that other people participate in climate activism could influence social norms, as well as the behavioral outcomes. With the current experimental setup, there is no way to rule this confounding variable out. I do

understand that it is important to provide participants with enough opportunities to actually carry out environmental activism behavior, and to guide them towards the climate groups that are working together with the researchers. However, without any control/ baseline phase which only includes the bulletin, there is no way to draw a clear causal interpretation of the results to the video intervention. Please see our answer to point 3 (Reviewer 2).

Validation of the intervention:

The current design might not allow to clearly identify all of the tested factors that may cause climate activism. Some might be not identified (or misidentified as being not important) due to the intervention not being sufficiently pilot-tested and validated.

It would make sense to first validate the video intervention and see whether all factors are influenced equally over the course of the 12 weeks. Then – after making sure that the intervention influences each factor comparably effectively – in a separate sample, information about activism events (bulletin) could be given and activism behavior could be assessed.

Without this validation, it is difficult to interpret the relative importance of the factors. For example, an increase in affective engagement could be found to predict an increase in activism behavior much better than all the remaining 10 factors. However, it is difficult to say whether this would be due to the intervention being particularly effective at manipulating affective engagement (and not really influencing the other 10 factors), or whether affective engagement actually is the most important predictor for activism. This concern should be addressed/ considered, and it should be communicated how the authors plan to interpret those results.

We would like to point the reviewer's attention to our acknowledgement that we are not testing a very specific theory about the relative weight of each factor in triggering activism (page 4). This, in fact, would be a next step after testing our design on a simpler theory: that one or more of the factors previously found to correlate with activism, if boosted, can cause objective participation. This study is aimed at verifying that our intervention is actually able to boost any of the 11 factors of interest. If the results are encouraging, we could then move on to test more complex theories of how increasing factor X by amount Y causes an increase in activism behavior, and how the 11 factors interact with each other. But testing the weight or interaction of the factors can only come after having developed and tested a design that is able to tackle causality between the factors and behavior. There is a great deal we need to figure out, and this N=143, 3 month longitudinal study, at an expense of over \\$15,000 in subject payments, should be able to get us there.

The current design can only provide insights on how effective the intervention was to manipulate the 11 factors and whether factors that have been effectively manipulated predict activism behavior.

4. Would the clarity and degree of methodological detail be sufficient to replicate exactly the proposed experimental procedures and analysis pipeline?

In order to fully understand the intervention (and replicate the study), more information about the content of the other 11 videos would be needed. For example, do the participants watch the same video every week? Was the first video (the example video) a general introductory video and the following 11 weeks will target each of the 11 identified factors separately? If so, was there a specific order chosen, based on ideas about how the factors might influence each other? Would the intervention work in a different order, too?

We have more clearly specified the details of the video intervention design (page 10, line 24)

“Video Intervention

Our video intervention is designed to boost the 11 psychological factors of interest. The intervention consists of 12 videos, each of which contains a mixture of two or more of the 11 psychological factors of interest. Each video is built on a theme (1. intro, 2. environmental threat, 3. human threat, 4. energy sources, 5. politics, 6. climate justice, 7. the climate movement, 8. victims and perpetrators, 9. neoliberalism and consumerism, 10. obstacles to engage, 11. how change happens, 12. imagine a climate-friendly world), and the 12 themes were selected based on the thematic curriculum of a college course on climate change taught by Dr. Aron. The choice

to build “thematic” videos, rather than building each video around on one of the 11 factors was done for two reasons: first, we are hoping to repeat the mobilizing effect of Dr. Aron’s class by adopting its thematic design which has repeatedly motivated students to climate action, and second, to prevent the participants from recognizing too easily in the videos the factors we are trying to boost.”

Additionally, we now have added two more example videos (Week 8 and Week 9 video).

Furthermore, more information about how the cooperating climate groups are collecting data about the attendance of participants would be desirable for replication. Additionally, in order to prevent potential privacy concerns about data collection (since the study did not seem to be reviewed by an ethics committee), more transparency would be great.

We have now added more details on how we plan on tracking attendance at the activism events (page 12, line 20)

“Activism behaviors. We will embed a study-team member in the partner organizations. She will monitor how many climate activism events are attended by each participant (which will be held, during this period, on Zoom, due to the ongoing covid-19 pandemic). She will do so by comparing all sign-in names on Zoom calls with the names of the participants of our study, and if any match is found, one “event score” will be added for each participant attending. This will be repeated at every event, so that for all participant of our experiment, there will be a record showing which and how many events they attended.”

5. Do the authors provide a sufficiently clear and detailed description of the methods to prevent undisclosed flexibility in the experimental procedures or analysis pipeline?

- The questions for the semistructured interviews seem to be adapted depending on the results of the quantitative data. If so, it would be good to have a little more transparency as to what questions are chosen and why/ depending on what results. Additionally, it would be great to be provided with the reasons for why only people who attended at least one climate activism event are considered for the interviews. This is a little confusing since an interest in investigating perceived barriers is expressed several times throughout the article.

We have now specified our strategy for selecting the final interview questions (page 12, line 55)

“With these general aims in mind, and learning from each participant’s quantitative results, we will tailor an interview. Possible questions could relate to changes, or lack of change, within a participant’s baseline- and follow-up surveys, as well as notably high or low engagement with climate action. For an example of what this interview schedule might look like, please see Appendix 1.”

- “If any composite score within each timepoint has a Cronbach’s alpha $< .5$, questions will be removed until the alpha is above this threshold. If there is no such solution, a single face-valid item will be chosen prior to hypothesis testing.” From my understanding, for at least some of the factors, previously validated scales are used. Removing questions to increase Cronbach’s alpha would undermine the practice of validating questionnaires (and therefore reduce the scale’s validity), especially if this results in only one face valid item left to be used.

The reviewer is correct; because the survey questions we will use to measure most of the factors have already been validated by previous studies, we don’t need to use the Chronbach’s alpha treatment. However, for Theory of Change, which is a novel factor for which we are using a novel scale, we will use the above Chronback alpha analysis in order to validate our novel scale. This is now specified (page 14, line 6)

“...However, we note that factor 11 (“Theory of Change) is novel. Therefore, we will do a reliability analysis. If the composite score within each timepoint for this factor has a Cronbach’s alpha $< .5$, questions will be removed until the alpha is above this threshold. If there is no such solution, a single face-valid item will be chosen prior to hypothesis testing.”

- Analytic plan H2: "If the subjective and objective event attendance is correlated at $r > .7$, the test with the subjective outcome will be dropped to reduce the number of tests." Does this mean, the plan is to run a paired sample t-test between baseline subjective and follow-up objective activism? (Since there is no measure of baseline objective activism?)

Having now added a baseline phase for objective activism (events attended), we will run a paired sample t-test between baseline objective and follow-up objective activism. While for the self-reported measures, both baseline and follow up activism will be self-reported (environmental education and leadership behaviors and emissions-reduction behaviors).

6. Whether the authors have considered sufficient outcome-neutral conditions (e.g. positive controls) for ensuring that the results obtained are able to test the stated hypotheses

Not really applicable here?

Reviewer: 3

Comments to the Author(s)

This pre-registered study proposes to identify the psychological factors that are most strongly associated with increased engagement in real-world collective climate action as a result of a 12-week intervention. Rather than testing a specific theory about the link between psychological factors and action, the authors propose two novel methodological tools: a video that attempts to boost 11 identified psychological factors and a behavior-tracking methodology to measure real-world collective action. The proposed single-cohort pre-post longitudinal study will recruit 160 participants aged 18-30 who will complete an online survey assessing the 11 psychological factors of interest, as well as self-reporting baseline activism behaviors and demographics. They will complete a 12-week video-based intervention, during which their activism will be monitored. At the end of the 12 weeks, they will complete the questionnaires for a second time. Proposed analyses will assess the pre-post change in the psychological factors, in behavioral outcomes (objectively monitored and self-reported), and the relationship between these changes.

The research question is timely and important and is well-motivated by the introduction. The methodology does a good job of outlining the design, the sample size, procedures related to outliers, and proposed analyses, including exploratory analyses - though some key gaps, including specification of DVs, need to be addressed, as outlined below. The issue of power is addressed. The proposed methodology and analysis pipeline appear largely sound and feasible, and are able to test the stated hypotheses, though I have some specific questions and concerns, which are outlined below. If these concerns are addressed, the project is likely to generate a rich and important dataset that will demonstrate whether video-based interventions can boost collective climate action, and will give new insights into the psychological factors that contribute to such change.

1. In the abstract and introduction, the authors highlight a lack of causal evidence regarding the psychological factors that predict collective climate action. A key drawback of the proposed single-cohort pre-post design, however, is that the **absence of a control group** will preclude causal inferences. While I sympathize with the arguments made regarding the sample size and cost entailed by the inclusion of a control group, the fact that the lack of a control group weakens claims to causal links should be acknowledged and, as a result the authors should perhaps rethink their focus on causal evidence. Please see the Baseline point we addressed from Reviewer 2 (point3). We have now taken the suggestion to add a Baseline period which serves as a control.

2. The 11 psychological factors that will be investigated are not well motivated in the introduction - there is general reference to psychological factors rather than specific reference to the 11 that will be examined here. The rationale for these specific 11 factors should be expanded.

We would like to point the reviewer's attention to our justification for the selection of the specific 11 factors (page 5, line 6)

“Although our review of the literature revealed more than a dozen possible psychological factors, we settled on just 11. These had both had the most correlational evidence with activism behavior, and fit our intuitions, as climate activists, of likely being relevant.”

3. Further, it is unclear how the questionnaire that assesses these 11 factors was developed - it appears that questions were selected from previous studies and instruments, but how this was done, and whether any effort has been made to pilot this instrument is not reported. There is very unequal treatment of the 11 factors in the questionnaire - each factor is assessed with a different number of items (e.g., some factors are assessed with three questions, some with as many as 16) and using different types of response scales (some -3 to 3; some 1 to 7). The variable psychometric characteristics could pose a significant confound for any attempt to identify the factors that contribute to action. It is not clear why there are such differences, what their potential impact is, and how they will be controlled for or any potential impact mitigated against - other than the proposal to exclude questions that reduce Cronbach's alpha or to select "a single face-valid item".

Because we decided to adopt scales that had been previously used in the literature to measure the respective factors, we also decided to abide by the previously used number of questions and score scales (whether 1 to 7 or -3 to +3) for each factor. In order to standardize these scores, we will z-score them and then take the average score for each subject for both baseline and follow-up surveys. Regarding the Cronbach's alpha analysis plan, see point 5 to Reviewer #2 above.

4. Further details on the video intervention are required. First - as is the case for the questionnaire, it is unclear how the video intervention was developed and whether any attempt has been made to validate it - for example to assess the extent to which the 11 psychological factors are present or evoked by the video. It is not clear whether the participants will see the same or a different video each week. If the video is different, how will the quality/extent to which these factors are targeted be controlled across weeks? Will order of presentation be counterbalanced across participants?

Please see point 4 from Reviewer 2.

5. Another shortcoming of existing research highlighted by the authors is the reliance on self-report or intentions rather than objectively measured participation - the authors proposed a novel "behavior-tracking methodology to measure real-world participation in climate action." This methodology is scantily described, however. It is not clear how the "partner organizations will monitor how many climate activism events are attended by each participant." This seems practically and ethically challenging to do and further details are needed. It seems likely that the authors will ultimately have to rely on self-reported participation, which again weakens the claims regarding novelty and advance on previous work. If the behavior-tracking methodology is successful, what will be measured, over what time scale, and how this will be included as an outcome variable in the analyses is not specified.

Please see the point 4. By Reviewer 2 above.

6. In the section on page 11 addressing outcome variables, climate activism events attended are not included as an outcome variable. As mentioned in the previous comment, it is not clear how the objective measure will be computed.

We have now corrected the outcome section (page 14, line 18)

“Outcomes. Objective participation in activism will be computed for each subject as the mean number of events attended (from the events bulletin) during the baseline period and during the intervention period. “

7. How will demand characteristics be addressed?

The participants will not have direct interactions with the experimenters (us), but rather watch videos asynchronously; they also are told that they will be paid at the end irrespective of the alerts (provided they simply read them). It is unlikely that participants would take hours of their lives to zoom into climate activist events [our embedded study team member can confirm if participants stay the duration of meetings] only to please a researcher they have never met; they are even less likely to do this since they would have no reason to think we could measure their attendance. Finally, the baseline control helps

somewhat; as does the fact that our study is really about which psychological factors predict activist behavior. Yes, demand characteristics could add noise, but we doubt they will 'work in favor' of any findings of correlations between psychological factors and activism.

8. The study has the potential to generate a very rich and valuable dataset - a further potential exploratory analysis might include factor analyses of the questionnaire data, which might aid in data reduction, and in the identification of the most important factors/families of factors.

We like that suggestion, but we'll keep it out of the report now. If we decide to perform this analysis later, we can report it as post-hoc findings.

Appendix C

Dear Editor,

We were very pleased of the final decision regarding the in-principle acceptance of our manuscript. We have adjusted the manuscript to reflect the reviewers' considerations in their latest comments. We have also done one last grammar check, following which some of the wording has been minorly changed. All modifications have been highlighted in yellow.

Again, we thank you and the reviewers for your efforts and for an enriching review process.

Sincerely,

Anna Castiglione, Adam Aron, Cameron Brick, Stefanie Holden and Debra Lindsay

Reviewer: 2

Comments to the Author(s)

After looking at the comments to the reviewers and the revised manuscript, I think the mentioned concerns have been addressed well.

I like how a baseline-phase was included, which helps with many original concerns and improves the overall quality of the study and the foundation of the causal interpretation. I was particularly impressed how the potential concern that participants attend in events other than the partner climate organizations' was anticipated and addressed. Several issues with transparency and the wording of the hypotheses have been addressed, as well.

The only thing that came to my mind reading the manuscript now was the exclusion of participants who attended more than one climate activism event. As mentioned before, I like the rationale for choosing this participant restriction. Nonetheless, I think it might be helpful to acknowledge/ mention more explicitly that only a somewhat specific sample of participants (resulting from the integration of the screening questions) will be collected (i.e. participants who believe in anthropogenic climate change but have not previously engaged in climate activism). I think this is an important subgroup to target (and maybe even the subgroup with the biggest potential for being mobilized), but it might be helpful to acknowledge that this is still a subgroup of the population regarding factors such as climate change beliefs and climate activism. However, this is no concern about methodological challenges but just an issue of wording.

We have added the following sentence to specify our exclusion criteria (page 7, line 8):

“Also, given our exclusion criteria (see screening survey), our study sample is limited to participants believing in climate change but having low or no prior engagement in climate activism. “

One other small point I noticed is in Page 7, Line 27, where the text says, “Participants will undergo an initial screening for age (18-30), native English, (...)”. Judging from the comments to reviewer #1, I assume that this might be a leftover from the previous version of the manuscript and the requirement for participants to have spoken English from childhood on ought to be removed?

We have now removed “native English” as a selection criteria.

Overall, the study sounds very interesting and I am excited about the commitment to study "actual" climate activism behavior instead of relying only on questionnaires. I am looking forward to learning about the results of the study!

Appendix D

Dear Editor,

Please find the attached manuscript submitted as a Stage 2 Registered Report: "Discovering the psychological building blocks underlying climate action: a longitudinal study of real-world activism".

On page 2 of the manuscript, we included a URL for the archived study data (raw and processed), the code that was used for the analyses, and a URL for the approved Stage 1 protocol stored in the PsyArXiv repository.

We confirm that no data for any pre-registered study other than pilot data included at Stage 1 was collected prior to the date of in principle acceptance (IPA). After receiving permission from the editor, we additionally ran an exploratory replication of the main study and report that in this Stage 2 manuscript, while carefully and explicitly separating the confirmatory Stage 2 study from that exploratory replication.

We thank you for the opportunity of advancing our manuscript to the Stage 2 of the preregistration process, and for considering our updated manuscript.

Sincerely,
The Authors

Appendix E

Discovering the psychological building blocks underlying climate action - a longitudinal study of real-world activism Royal Society Open Science

This manuscript deals with a very important and pressing issue – climate activism. I furthermore think the study makes a meaningful contribution to the literature by using and suggesting an interesting way to measure actual activism behaviour, as well as reporting interesting results. The introduction, rationale, and stated hypotheses are the same as the approved stage 1 submissions and the authors adhered precisely to the registered experimental procedures and explicitly stated whenever they included unregistered information or analyses. The authors conclusions are overall justified. However, I recommend some revisions involving minor edits, as well as revisions regarding how the statistical analyses are reported, and the accessibility of the data.

Revisions:

Page 6, paragraph 2, sentence 2: “(such as circulation of petitions, raising awareness about climate issues, outreach and community organizing)(Bamberg et al., 2018; Barkan, 2004; Lubell, 2002; Ockwell et al., 2009; Rees & Bamberg, 2014; Seguin et al., 1998).”

To my knowledge, citations should be separated from parenthetical text with semicolons: “(such as circulation of petitions, raising awareness about climate issues, outreach and community organizing; Bamberg et al., 2018; Barkan, 2004; Lubell, 2002; Ockwell et al., 2009; Rees & Bamberg, 2014; Seguin et al., 1998).” This applies throughout the paper.

Page 7, paragraph 1, sentence 1: “...and other countries”

I wonder if it makes sense to spell out what other countries these findings emerged in. This would be especially interesting if these countries are not predominantly white, developed western nations.

Page 17, paragraph 1, sentence 1: “...Following the confirmatory analyses,...”

When exactly were the semi-structured interviews conducted? It would be interesting to know how much time passed between the intervention and the interviews.

Page 20, Paragraph 1, sentence 2 & 3: “However, we note that factor 11 (theory of change) was novel. Cronbach's $\alpha = .56$ for theory of change at both time points.”

The second sentence may better be combined with the first one. Additionally, Cronbach's alpha for theory of change is reported in the text as .64, however, in the supplementary materials it is reported as .63

Page 21, paragraph 1, sentence 6: “We adhered to the guidelines of thematic analysis (Braun & Clarke, 2006).”

This citation seems to be differently formatted than the rest of the text (bold & smaller font size).

Page 21, paragraph 2: Post- hoc follow-up survey

The paper would flow better and lead to less confusion if this part was explained between the semi-structured interview and the exclusion criteria.

Page 21, paragraph 3: “Three psychological factors significantly changed from pre to post intervention”

This sentence needs specification whether these three factors have *increased* (or decreased) in order to fit to the hypothesis stated above. Furthermore, t-values and degrees of freedom should be reported. An additional table (e.g., in the supplementary materials) with the mean values and standard deviations for all factors/ behaviours pre and post intervention would be helpful (not only the change scores).

Page 22, H2:

Non-significant t-tests should still be reported. Can be combined in a summary statistic (e.g., all $t(df) < XYZ$, all $p > XYZ$, all $d < XYZ$).

Page 25, Paragraph 2, sentence 4:

Like above, t-values and degrees of freedom are missing.

Page 26, Unregistered Pilot study: “This was an in-person, six-week replication of Study 1 and did not include a baseline. The aim was to test whether the in-person format increased activist participation.”

I understand that a baseline period was not very feasible for study 2. However, the claim in the second sentence should be adapted accordingly: Without a baseline period or control group it does not seem possible to *test* whether the format *increased* activist *participation*. The same logic applies to the hypothesis in page 28 (first sentence). If I understood the study design correctly, it can be tested whether the in-person intervention increased the 11 psychological factors and self-reported climate-related behaviours. In addition, a regression analysis can reveal whether the increase in the 11 factors predicts attendance in activist events. But it does not seem possible to test whether the intervention increased event attendance.

Page 27, Paragraph 1, sentence 1:

Rationale for choice of sample size is unclear considering the results from the power analysis in study 1 (N=143).

Page 28, Results, sentence 1:

Like above, t-values and degrees of freedom are missing. Results for self-reported climate related behaviours seem to be missing and would have been very interesting.

Page 28, paragraph 3, last sentence: “We suggest that including a social component, in-person study events, and activist events is not sufficient to *increase* attendance at activist events,...”

Same logic as above.

Page 28 & Page 29, General Discussion, paragraph 1 & 2: Affective engagement vs. Risk perception

Affective engagement and risk perception → only one term should be chosen for the discussion and used continuously, unless talking about the difference between both.

Data:

The accessibility of the data could be improved by providing a dataset that combines pre- and post scale values/ means of the relevant factors and dependent variables for each participant. It would be furthermore advisable to double-check the description of the measures in the supplementary materials and compare it with the available datasets, as there is some conflicting information (e.g., the supplementary materials say there are four items for the Self-Efficacy scale, but the datasets include 6 SE items).

Appendix F

Dear Editor,

We adjusted the Stage 2 manuscript to reflect the reviewers' latest comments. We also made small, cosmetic changes to the Stage 1 sections for clarity. All modifications are highlighted in yellow.

Thank you and the reviewers for your time and help in significantly bettering the manuscript.

Reviewer: 1

1) Comments to the Author(s)

The manuscript is sound, my only comment is that the authors claim to have completed a thematic analysis in study, however, the qualitative analysis appears to be a summative content analysis not a thematic analysis. I recommend the authors update how they have labelled the analysis. Here is a suggested paper for describing content analysis.

<https://journals.sagepub.com/doi/abs/10.1177/1049732305276687>

Thanks for this correction. We changed the labeling of the qualitative analysis to a "content analysis." This change occurs on page 20 and is also reflected in the citation of Elo and Kyngas, 2008.

Reviewer: 2

Comments to the Author(s)

This manuscript deals with a very important and pressing issue – climate activism. I furthermore think the study makes a meaningful contribution to the literature by using and suggesting an interesting way to measure actual activism behaviour, as well as reporting interesting results. The introduction, rationale, and stated hypotheses are the same as the approved stage 1 submissions and the authors adhered precisely to the registered experimental procedures and explicitly stated whenever they included unregistered information or analyses. The authors conclusions are overall justified. However, I recommend some revisions involving minor edits, as well as revisions regarding how the statistical analyses are reported, and the accessibility of the data.

Revisions:

- 2) Page 6, paragraph 2, sentence 2: "(such as circulation of petitions, raising awareness about climate issues, outreach and community organizing)(Bamberg et al., 2018; Barkan, 2004; Lubell, 2002; Ockwell et al., 2009; Rees & Bamberg, 2014; Seguin et al., 1998)." To my knowledge, citations should be separated from parenthetical text with semicolons: "(such as circulation of petitions, raising awareness about climate issues, outreach and community organizing; Bamberg et al., 2018; Barkan, 2004; Lubell, 2002; Ockwell et al., 2009; Rees & Bamberg, 2014; Seguin et al., 1998)." This applies throughout the paper.

We corrected this issue as the reviewer suggested.

- 3) Page 7, paragraph 1, sentence 1: "...and other countries" I wonder if it makes sense to spell out what other countries these findings emerged in. This would be especially interesting if these countries are not predominantly white, developed western nations.

We agree and now list additional Middle Eastern and Global South countries of some of the climate beliefs/action analyses: "... countries such as Argentina, Chile, South Korea, Mexico, Russia, Turkey and others; Hornsey et al., 2016" on page 6.

- 4) Page 17, paragraph 1, sentence 1: "...Following the confirmatory analyses,..." When exactly were the semi-structured interviews conducted? It would be interesting to know how much time passed between the intervention and the interviews.

On page 15, we now specify that the interviews were carried out within two months from the end of the study: “Following the confirmatory analyses, we conducted semi-structured interviews within two months from the end of the study...”

- 5) Page 20, Paragraph 1, sentence 2 & 3: “However, we note that factor 11 (theory of change) was novel. Cronbach's $\alpha = .56$ for theory of change at both time points.” The second sentence may better be combined with the first one. Additionally, Cronbach's alpha for theory of change is reported in the text as .64, however, in the supplementary materials it is reported as .63

Thanks for pointing out these errors. We have combined the two sentences on page 20: “However, we note that factor 11 (theory of change) was novel, and Cronbach's $\alpha = .56$ for this factor, at both time points. To reach the minimum reliability score of $\alpha = .6$, one item was dropped, which led to $\alpha = .64$.”

- 6) Page 21, paragraph 1, sentence 6: “We adhered to the guidelines of thematic analysis (Braun & Clarke, 2006).” This citation seems to be differently formatted than the rest of the text (bold & smaller font size).

Thanks: we corrected this formatting. The citation changed due to another reviewer comment below.

- 7) Page 21, paragraph 2: Post- hoc follow-up survey. The paper would flow better and lead to less confusion if this part was explained between the semi-structured interview and the exclusion criteria.

We thank the reviewer for this suggestion, but RSOS guidelines require post hoc analysis be placed at the end of the study description.

- 8) Page 21, paragraph 3: “Three psychological factors significantly changed from pre to post intervention.” This sentence needs specification whether these three factors have *increased* (or decreased) in order to fit to the hypothesis stated above. Furthermore, t-values and degrees of freedom should be reported. An additional table (e.g., in the supplementary materials) with the mean values and standard deviations for all factors/behaviours pre and post intervention would be helpful (not only the change scores).

We made the above corrections and additions on page 20: “Three psychological factors significantly increased from pre to post intervention (after applying Bonferroni-Holm correction): affective engagement $t(95) = 3.25, p = .02, d = .33$, collective efficacy $t(95) = 4.94, p < .001, d = .50$, and self-efficacy $t(95) = 4.73, p < .001, d = .48$ (Fig.2).”

- 9) Page 22, H2: Non-significant t-tests should still be reported. Can be combined in a summary statistic (e.g., all $t(df) < XYZ$, all $p > XYZ$, all $d < XYZ$).

We now report the summary statistics for the non-significant t-tests on pages 20 and 21 (study 1) and 27 (study 2):

Page 20: “No other factors changed, all $t(95) < 2.50$, all $p > .17$ all $d < .26$).”

Page 22: “self-reported environmental education and leadership had no change (-0.02 out of 7), and self-reported emissions reduction also had no change (+0.01 out of 7), all $t(95) < 1.00$, all $p > .99$ all $d < .10$).”

Page 27:” None of the other factors changed all $t(37) < 2.97$, all $p > .05$, all $d < .49$).”

Page 27: “Self-reported activist behavior did not increase, neither for environmental education and leadership (+0.09 out of 7), nor for emissions reduction behaviors (+0.21 out of 7) (all $t_s(37) < 1.60$, all $p_s > .10$ all $d_s < .30$).”

10) Page 25, Paragraph 2, sentence 4: Like above, t -values and degrees of freedom are missing.

We added the t and df values on page 27: “Two psychological factors increased after the intervention (after Bonferroni-Holm correction): self-efficacy $t(37) = 3.62$, $p = .01$, $d = .33$ and identity $t(37) = 3.60$, $p < .001$, $d = .59$. None of the other factors changed all $t_s(37) < 2.97$, all $p_s > .05$, all $d_s < .49$).”

11) Page 26, Unregistered Pilot study: “This was an in-person, six-week replication of Study 1 and did not include a baseline. The aim was to test whether the in-person format increased activist participation.” I understand that a baseline period was not very feasible for study 2. However, the claim in the second sentence should be adapted accordingly: Without a baseline period or control group it does not seem possible to *test* whether the format *increased* activist *participation*. The same logic applies to the hypothesis in page 28 (first sentence). If I understood the study design correctly, it can be tested whether the in-person intervention increased the 11 psychological factors and self-reported climate-related behaviours. In addition, a regression analysis can reveal whether the increase in the 11 factors predicts attendance in activist events. But it does not seem possible to test whether the intervention increased event attendance.

Attendance of this population at those meetings was verifiably almost zero, so any attendance would be an increase. Nonetheless, to avoid confusion we changed the “increase” verb to “trigger” at the bottom of pages 26 and 27, and on page 28.

12) Page 27, Paragraph 1, sentence 1: Rationale for choice of sample size is unclear considering the results from the power analysis in study 1 ($N=143$).

On page 25 we added the following statement: “This small sample (smaller than the N prescribed by our power analysis for Study 1) seemed appropriate for a small pilot study with no planned comparison between two groups (experimental and baseline).”

13) Page 28, Results, sentence 1: Like above, t -values and degrees of freedom are missing. Results for self-reported climate related behaviours seem to be missing and would have been very interesting.

We added the t and df values and the results for self-reported climate related behaviors on page 27: “Self-reported activist behavior did not increase, neither for environmental education and leadership (+0.09 out of 7), nor for emissions reduction behaviors (+0.21 out of 7) (all $t_s(37) < 1.60$, all $p_s > .10$ all $d_s < .30$).”

14) Page 28, paragraph 3, last sentence: “We suggest that including a social component, in person study events, and activist events is not sufficient to *increase* attendance at activist events,…” Same logic as above.

See changes in #11 above.

15) Page 28 & Page 29, General Discussion, paragraph 1 & 2: Affective engagement vs. Risk perception. Affective engagement and risk perception □ only one term should be chosen for the discussion and used continuously, unless talking about the difference between both.

We edited the discussion to only mention "emotional engagement" (and not risk perception), given that this distinction was not part of the initial analytical plan.

- 16) Data: The accessibility of the data could be improved by providing a dataset that combines pre- and post scale values/ means of the relevant factors and dependent variables for each participant. It would be furthermore advisable to double-check the description of the measures in the supplementary materials and compare it with the available datasets, as there is some conflicting information (e.g., the supplementary materials say there are four items for the Self-Efficacy scale, but the datasets include 6 SE items).

We created the merged dataset and checked that the scales reported in the supplementary data match the result datasets.

Reviewer: 3

Comments to the Author(s)

The paper by Castiglione et al. is a Stage 2 Registered report. The authors have done an excellent job of implementing and reporting on the study as proposed, as well as conducting an important exploratory replication. While the results were disappointing, the study provides valuable insights that will benefit future research. In particular, the analysis of semi-structured interview data to understand what held back the participants from engaging in the activism events, and the lessons emerging from the in-person replication, which also failed to increase engagement in activism, highlight the challenge of fostering collective action and will stimulate further work.

- 17) Regarding the results - a fuller presentation of the data and results of analyses carried out for H1 and H2 is required. Ideally, a table should provide means and standard deviations, t-statistics, p-values and effect sizes for each of the 11 psychological factors and three behavioural outcomes examined. Alternatively, the data could be presented in graphical form, as in Figure 2 (though please see comment below).

As mentioned in the answer to Reviewer 2, we added a table reporting all results statistics for the 11 factors and 2 self-reported behaviors. For objective participation statistical analysis is not necessary since the results were clear: only 3 out of 97 participants joined only a couple of events.

- 18) A second comment is that I found the general discussion very short. The brief discussion seems like a missed opportunity to engage with the question of how the successfully induced increases in affective engagement and self- and collective-efficacy (with the latter two theorised to constitute key factors in individual and collective action) might be converted into active participation. A second important observation was the fact that the pressures of modern student life (juggling college-work plus work-work) form a significant barrier to engagement in activism, despite an awareness of and concern about climate. How to overcome the near-universal barrier of time poverty may be one of the most important questions we should examine.

Thank you for these useful suggestions. We added two paragraphs addressing the two points suggested by the reviewer.

Page 28: " Only four of the eleven factors changed in either study, suggesting that changes this size in these four factors alone were not sufficient to trigger action. Re-designing the videos might help to boost all 11 factors. Future research could employ a professional videographer, create content that is more local to the participants, and use forms of narrative structure, perhaps by involving actors. Or, additional factors might need to be discovered and targeted. It remains possible that large increases in the 11 factors might not be

sufficient to trigger action when participants do not have the time or financial resources to volunteer for activism.”

Page 29:“ One potential incentive to join action for those lacking time or financial resources could be the support system that is often created inside an activist group—something our study did not encourage. Our personal observations point to emotional and social benefits of participation. For example, when activists share values and struggles inside the activist space, they become empathetic with each other and can learn solidarity skills of mutual aid and support to overcome engagement obstacles. For example, they can cover each other’s work shifts to allow one to go to a rally or help each other find jobs within activism or advocacy. These connections with other activists also lead to valuable professional networks. Accordingly, interventions might trigger more engagement by advertising these benefits or designing the intervention to foster these exchanges. We also recommend providing more event times that fit busy schedules, and specifically recruiting participants already interested or engaged in climate activism.”

19) Minor comment - in Figure 2, a better way to present the data would be to use separate graphs for each psychological factor and to present "post" and "pre" side-by-side (i.e., use hue for timepoint, not factor).

We edited Figure 2 as the reviewer suggested: